



# Measurement report: Vertical distribution of biogenic and anthropogenic secondary organic aerosols in the urban boundary layer over Beijing during late summer

Hong Ren[1,2], Wei Hu[1], Lianfang Wei[2], Siyao Yue[2,a], Jian Zhao[2], Linjie Li[2,b], Libin Wu[1], Wanyu Zhao[2],
Lujie Ren[1], Mingjie Kang[2], Qiaorong Xie[1,2], Sihui Su[1], Xiaole Pan[2], Zifa Wang[2], Yele Sun[2], Kimitaka
Kawamura[3], and Pingqing Fu[1]

[1] Institute of Surface-Earth System Science, School of Earth System Science, Tianjin University, Tianjin 300072, China
[2] State Key Laboratory of Atmospheric Boundary Layer Physics and Atmospheric Chemistry, Institute of Atmospheric
Physics, Chinese Academy of Sciences, Beijing 100029, China
[3] Chubu Institute for Advanced Studies, Chubu University, Kasugai 487-8501, Japan
[a] Now at: Minerva Research Group, Max Planck Institute for Chemistry, Mainz 55128, Germany
[b] Now at: Department of Chemistry and Molecular Biology, University of Gothenburg, 412 96, Gothenburg, Sweden

**Correspondence**: Pingqing Fu (fupingqing@tju.edu.cn)

**Abstract.** Secondary organic aerosols (SOA) play a significant role in atmospheric chemistry. However, little is known
about the vertical profiles of SOA in the urban boundary layer (UBL). This gap in the knowledge constrains the SOA
simulation in chemical transport models. Here, we synchronously collected aerosol samples at 8 m, 120 m and 260 m based
on a 325-m meteorological tower in urban Beijing from August 15[th] to September 10[th], 2015. Strict emission controls were
implemented during this period for the 2015 China Victory Day Parade. The sum of biogenic SOA tracers increased with
height. The fraction of SOA from isoprene oxidation increased, whereas the fraction of monoterpene and sesquiterpene SOA
decreased with height. The 2,3-dihydroxy-4-oxopentanoic acid (DHOPA), one tracer of anthropogenic SOA from toluene
oxidation, also increased with height. The complicated vertical profiles of SOA tracers highlighted the needs to measure
SOA within the UBL. The sum of estimated secondary organic carbon (SOC) ranged from 341 to 673 ngC m$^{-3}$. The increase
in the SOC fraction from isoprene and toluene with height was found to be more related to regional transport whereas the
decrease in the SOC from monoterpene and sesquiterpene with height was more subject to local emissions. Emission
controls during the parade reduced SOC by 4–35% with toluene SOC decreasing more than the other SOC. This study
demonstrates that vertical distributions of SOA within the UBL are complex, and the vertical profiles of SOA concentrations
and sources should be considered in the future field and modelling studies.

## 1 Introduction

In the middle of the 20[th] century, atmosphere pollution events began to be frequently reported in different regions worldwide
(Went, 1960; White and Roberts, 1976; Barrie, 1986; Huang et al., 2020a). Subsequently, many studies on atmospheric
aerosols were undertaken in order to understand the sources and evolution mechanisms of aerosols and discuss their effects



on climate and human health. It is well known that atmospheric aerosols can impact on radiation forcing, the hydrological cycle, regional and global climate and human health (Ramanathan et al., 2001; Kanakidou et al., 2005; Pöschl, 2005; Li et al., 2016; Wang et al., 2016; Shrivastava et al., 2017a; Hu et al., 2020; Su et al., 2020). Generally, atmospheric organic aerosols (OA) contribute 20–90% of the particulate matter (PM), in which ca. 30–70% are secondary organic aerosols (SOA)

(Kanakidou et al., 2005; Jimenez et al., 2009; Ervens et al., 2011; Huang et al., 2014). SOA is generally formed through the photooxidation a of volatile organic compounds (VOCs), including biogenic VOCs (BVOCs, e.g. isoprene, monoterpene, sesquiterpene and oxygenated hydrocarbons) from terrestrial vegetation and marine phytoplankton and anthropogenic VOCs (AVOCs, e.g. toluene and naphthalene) from biomass burning, coal combustion, vehicle exhausts and solvent use. Global models generally consider anthropogenic SOA (ASOA) accounts for a small fraction of SOA yield, and about 90% of

SOA is attributed to biogenic SOA (BSOA) (Kanakidou et al., 2005; Volkamer et al., 2006). BSOA and ASOA fractions are potentially under predicted in models according to the previous studies (Volkamer et al., 2006; Shrivastava et al., 2017b). In recent years, a large number of studies based on field observations suggested that the formation of BSOA can be enhanced by anthropogenic precursors, an effect which is known as anthropogenic-biogenic interactions (Goldstein et al., 2009; Shilling et al., 2012; Zelenyuk et al., 2017). Simultaneously, SOA can be transported at a regional to global scale,

influencing the climate by constraining radiation forcing and changing cloud condensation nuclei (CCN) size, and threatening human health (Pöschl, 2005; Russell and Brunekreef, 2009; Shrivastava et al., 2017b).

In the last decade, the severe haze air pollution in China resulting from rapid industrialization and urbanization has attracted worldwide attention (Fu and Chen, 2017; An et al., 2019; Huang et al., 2020a). The haze episodes in China are suggested to result from a complex interplay of anthropogenic emissions, atmospheric processes, regional transport, meteorological

conditions and climatic conditions (Wang et al., 2006; He et al., 2013; Guo et al., 2014; Zheng et al., 2015; Sun et al., 2016; Huang et al., 2018; An et al., 2019; Huang et al., 2020b; Du et al., 2021). The high contribution of secondary aerosols to the PM pollution during haze events in China, highlighting the urgent need to understand the compositions and processes of SOA in the atmosphere (Huang et al., 2014; An et al., 2019). Previous studies have reported the chemical characteristics of OA in many areas of China (Simoneit et al., 1991; Wang et al., 2006; Wang et al., 2012; Wang et al., 2013; Ren et al., 2016;

Yang et al., 2016; Li et al., 2018; Tang et al., 2018; Xie et al., 2020). However, the studies detailing the vertical properties of SOA in the urban boundary layer have not been widely reported, which constrains research on the interactions of aerosols and regional transport, local emissions, atmospheric processes and meteorological conditions in urban areas (Li et al., 2017). Vertical profiles of atmospheric structures, gases concentrations, bulk chemical compositions and nitrogen isotopes in the urban boundary layer (UBL) have been investigated in field campaigns (Chan and Kwok, 2000; Chan et al., 2005; Guinot et

al., 2006; Mao et al., 2008; Sun et al., 2015; Zhao et al., 2017; Sun, 2018; Wu et al., 2019). SOA observations were carried out on the Amazon Tall Tower Observatory (ATTO) in a rain forest region in the central Amazon Basin. The vertical gradients of VOCs in and above the forest canopy were measured, but this did not result in a complete vertical SOA profile (Andreae et al., 2015; Yáñez-Serrano et al., 2018). A previous study reported that SOA occurred more abundantly above the surface layer during the summer over the southeastern United States, which was potentially related to the heterogeneous





chemical reactions of BVOCs oxidation products and gas-to-particle reactions of semi-volatile VOCs oxidation products (Goldstein et al., 2009). This highlights the pressing needs to obtain the vertical SOA profiles in the cities, especially in a Chinese megacity frequently enduring serve air pollution. This would help in estimating SOA feedback under the control of local processes, regional transport and mixing of heights in urban areas, finding the predominant compounds or processes for

the formation of air pollution and improving the modeling of SOA in chemical transport models. This information also has regulatory implications for decision markers.

Beijing, one of the super megacities of China, held the 2015 China Victory Day parade in the late summer 2015. The government had implemented strict emission controls in Beijing and its seven surrounding provinces to guarantee the air quality. This provided a unique chance to study the chemical behaviours and regional transport of atmospheric aerosols

under the government interventions. Daily $PM_{2.5}$ samples were synchronously collected at three heights (8 m, 120 m and 260 m, above ground level, respectively) based on a 325-m meteorological tower in urban Beijing during the period 15th August to 10th September 2015. Observations at 8 m are more subject to local emissions whereas those at 120 m and 260 m are more representative mixing and/or regional scale influences (Sun et al., 2015; Zhao et al., 2020). BSOA and ASOA tracers in $PM_{2.5}$ were quantified by gas chromatograph/mass spectrometry (GC/MS); organic carbon (OC), elemental carbon (EC) and

water-soluble organic carbon (WSOC) in $PM_{2.5}$ were also determined. In addition, the tracer-based method (Kleindienst et al., 2007; Fu et al., 2014) was used to estimate the contributions of biogenic SOC (BSOC) and anthropogenic SOC (ASOC). The constraints of joint regional prevention and control implemented by the policymakers on SOC were also analyzed. To be best of our knowledge, this was the first time that vertical profiles of SOA tracers were measured at a molecular level in a megacity of China. This campaign provided new insights into the formation mechanisms of SOA in haze episodes and the

feedback on SOA influenced of local emissions, regional transport and mixing of heights over the North China Plain (NCP). Furthermore, this study provided a scientific basis for China's initiatives to guarantee air quality in Beijing and contributed to improving the simulations of SOA in the chemical transport models.

## 2 Materials and methods

### 2.1 Sampling

Daily $PM_{2.5}$ samples were collected at three heights: 8 m (at the rooftop of a two-story building near by the 325-m meteorological tower), 120 m and 260 m (at the platforms of the tower) in urban Beijing during the China Victory Day parade period (August 15th–September 10th, 2015). The sampling site is located in the typical urban site (between 3- and 4-ring of Beijing), which is surrounded by transport roads, public park, restaurants, residential housing and a gas station. Three sampling periods were classified according the date of implementing of emission controls by the government, which were

marked as before parade (Before-P): August 15th–19th; during parade (During-P): August 20th–September 3rd; and after parade (After-P): September 4th–10th, respectively. The aerosol sampling was conducted during 08:00–06:00 (local time) with pre-combusted (450 °C combusted for 6 h) quartz fibre filters (Pallflex, 8×10 in) using high-volume air samplers



(TISCH, USA) at a flow rate of 1.1 m$^3$ min$^{-1}$. Samples were placed in aluminium paper and kept in a refrigerator at –20 ºC in darkness until analysis.

Meteorological parameters including wind speed (WS), wind direction (WD), temperature (Tem), relative humidity (RH) at the heights of 8 m, 120 m and 260 m were measured based on the 325-m meteorological tower during the sampling period.

## 2.2 Carbonaceous component Analyses

OC and EC in an aliquot filter were directly analyzed by using an OC/EC Carbon Aerosol Analyzer (Sunset Laboratory Inc., USA) following a NIOSH protocol. The measuring method is described in detail in a previous study (Mkoma et al., 2013). An aliquot filter of 3.14 cm$^2$ was extracted with 15 ml ultrapure water under ultrasonication with ice-water for 20 min. WSOC in this water extract was determined with a total organic carbon (TOC) analyzer (Model NPOC, Shimadzu, Japan). The concentrations of OC, EC, and WSOC were calibrated with field blank filters.

## 2.3 Measurement of OA molecular compositions using GC/MS

A filter was extracted three times with dichloromethane/methanol (2:1, v/v) under ultrasonication. The extracts were then filtered, concentrated by a rotary evaporator and blown down to dryness. After that, the dried extracts were reacted with 60 μl of N,O-*bis*-(trimethylsilyl)trifluoroacetamide (BSTFA) with 1% trimethylsilyl chloride and 10 μl of pyridine at 70 ºC for 3 h. After sufficient reaction, 40 μl internal standard solvent (C$_{13}$ *n*-alkane, 1.43 ng μl$^{-1}$) was added to derivatives before GC/MS analyses. Three field blank filters were treated as real sample and used for quality calibration. GC/MS is performed on a Hewlett-Packard model 7890A GC coupled to Hewlett-Packard model 5975C mass selective detector (MSD). GC separation is equipped with a spitless injection and a fused silica capillary column (DB-5MS, 30 m×0.25 mm i.d., 0.25 μm film thickness). The GC oven temperature program was set as follows: 50 ºC hold 2 min, then increased to 120 ºC at a rate of 15 ºC min$^{-1}$, heated up to 300 ºC at a rate of 5 ºC min$^{-1}$, and finally hold at 300 ºC for 16 min. The Mass Spectrometer was operated on the electron impact (IE) mode at 70 eV and scanned from 50 to 650 Da. Organic marker measurements were determined by comparing with references, library and authentic standards, and were quantified with GC/MS response factors acquired using authentic standards or surrogates (Simoneit et al., 1991; Fu et al., 2008; Fu et al., 2009). The data reported in this work was corrected for the field bank but not for recoveries.

## 2.4 Air mass backward trajectory

In order to investigate the influences of air mass origins on urban aerosols of Beijing, 3-day backward trajectories starting at 300 m (a.g.l.) of every 6 hours were calculated for each sample using the HYSPLIT4 model (http://ready.arl.noaa.gov/HYSPLIT.php). Cluster analyses were applied to better estimate the contributions of air mass from each origin during the sampling period. As showed in Figure S1, total seven clusters were calculated with air clusters from south, southeast and northeast of Beijing accounting for more than 70%, suggesting the collected samples were most influenced by air masses from these directions.





Simultaneously, to better understand the potential pollution sources during the pollution events, the retroplumes of air masses for pollution days were also calculated by using the FLEXPART (FLEXible PARTicle dispersion) model (Figure S2). Detail information about the model was shown in previous studies (Brioude et al., 2013; Wei et al., 2018a) and the website (https://www.flexpart.eu). The model was set with a height of 300 m and three-day backward trajectories. The results

demonstrated that pollution days were largely impacted by the air masses from southern regions of Beijing.

## 3 Results and discussion

Meteorological conditions at the sampling site during the observation period are shown in Figure 1. Additional information has been reported in the previous publication (Zhao et al., 2017). The prevailing winds changed from easterly or westerly to northerly between 8 m and 260 m. Much lower wind speeds and less southern winds at the ground surface (8 m) during the

campaign were related to the influences of surrounding buildings. Vertical differences of wind speeds and directions suggest that samples collected at 8 m are more related to local sources emissions whereas samples collected at upper layers are more influenced by the regional scale. Ambient temperature decreased slightly versus relative humidity (RH) increased slightly with the height, which possibly plays a role on the vertical profile of the gas-to-particle partitioning of organic aerosols (Sun et al., 2015).

During the sampling period, three pollution episodes (marked as E1, E2 and E3) were recorded according to the pollution levels. The prevailing winds during these episodes varied with the height between 8 m and 260 m (Figure S3). The wind in the upper layers (120 m and 260 m) were mainly from the south whereas the wind in the ground surface layer (8 m) was from the north. Similar to the air mass footprints (Figure S2), these results suggest that the air masses from the southern region significantly contributed to the haze pollution in Beijing (Zheng et al., 2015; Tian et al., 2019).

The concentrations of WSOC and OC increased slightly with height ($2.73 \pm 1.31$ µgC m$^{-3}$ and $5.03 \pm 2.28$ µgC m$^{-3}$ at 260 m, $2.69 \pm 1.55$ µg m$^{-3}$ and $5.32 \pm 2.88$ µg m$^{-3}$ at 120 m, and $2.03 \pm 0.99$ µg m$^{-3}$ and $4.37 \pm 1.69$ µg m$^{-3}$ at 8 m, respectively) (Figure S4 and Table S1). The average concentrations of WSOC and OC at three heights showed no significant differences (Table S2), indicating that aerosols were well mixed within the boundary layer. However, the ratios of WSOC to OC at 8 m differed from the ratios in the upper layers (120 m and 260 m), which suggests the different oxidation processes or sources of

SOA and highlights the importance of investigating the vertical profiles of SOA in the UBL. Higher ratios of WSOC to OC were recorded in the upper layers than in the ground surface layer (Table S1), suggesting that organic aerosols in the upper layers were more oxidized than in the ground surface layer. In addition, primary sources from local dust and soil resuspension, such as primary biological aerosols which contain high abundance of water-insoluble organic compounds (WIOC), were potentially attributed to the higher fractions of WIOC to OC at the ground surface than at the upper layers

(Wang et al., 2019). Especially, correlation coefficient values ($R^2$) between WSOC and OC were 0.96, 0.93 and 0.47 at 260 m, 120 m and 8 m, respectively (Figure S5), suggesting a predominant contribution of secondary sources to OA in the upper layers.



Concentrations of identified secondary organic compounds are shown in Table S1, including BSOA tracers (isoprene, monoterpene and sesquiterpene oxidation products), ASOA tracers (DHOPA and phthalic acid for toluene and naphthalene oxidation products, respectively), poly acids and aromatic acids in the aerosols at three heights. Most of these molecular tracers showed higher abundance at high layers (≥ 120 m) than at 8 m, except for monoterpene and sesquiterpene SOA

tracers. Table S2 shows significant differences in the average concentrations of these SOA tracers with height, except for monoterpene SOA, which highlighted the needs to investigate the vertical distributions of SOA within the UBL. Many factors can regulate the vertical profiles of SOA: (1) lower temperature and higher RH in the upper layers than in the ground surface layer are potentially favourable to the condensation of semi-volatile organic compounds onto particles (Carlton et al., 2009; Hallquist et al., 2009); (2) local emission, regional transport and the mixing of heights can influence the loading of

SOA in aerosols (Brown et al., 2013); (3) Other factors, such as atmosphere oxidation capacity can play a role in the formation of SOA (Wang et al., 2018a). Thus, vertical distributions of SOA tracers provided useful information on the aerosol chemistry and the simulation of SOA in the urban boundary layer.

### 3.1 Vertical characteristics of SOA tracers

### 3.1.1 Vertical distribution of BSOA tracers

The total concentrations of BSOA tracers were $31.5 \pm 16.8$ ng m$^{-3}$, $36.4 \pm 26.1$ ng m$^{-3}$ and $50.2 \pm 27.0$ ng m$^{-3}$ at 8 m, 120 m and 260 m, respectively (Table S1). The increase in concentration with height is potentially linked to the regional transport and gas-to-particle processes of semi-volatile VOCs due to lower temperatures at the upper layers (Goldstein et al., 2009). Moreover, vertical convection transport and BVOCs emission sources cannot be ignored (Ran et al., 2012; Wei et al., 2018b). The total concentrations of isoprene SOA tracers were $19.7 \pm 12.0$ ng m$^{-3}$, $27.1 \pm 22.4$ ng m$^{-3}$ and $38.7 \pm 24.1$ ng m$^{-3}$ at 8 m,

120 m and 260 m, respectively, among which $C_5$-alkene triols (the sum of cis-2-methyl-1,3,4-trihydroxy-1-butene, 3-methyl-2,3,4-trihydroxy-1-butene and trans-2-methyl-1,3,4-trihydroxy-1-butene) were the most abundant compounds, followed by 2-methylerythritol (2-MET), 2-methylthreitol (2-MT) and 2-methylglyceric acid (2-MGA) (Table S1). The total of monoterpene SOA tracers were $10.5 \pm 5.18$ ng m$^{-3}$ (8 m), $8.45 \pm 3.68$ ng m$^{-3}$ (120 m) and $10.5 \pm 3.86$ ng m$^{-3}$ (260 m), with pinonic acid (PNA) being the most abundant at 8 m whereas 3-methyl-1,2,3-butanetricarboxylic acid (MBTCA) was the

dominant compounds at 120 and 260 m. Sesquiterpene SOA tracer (β-caryophyllinic acid) concentrations were $1.32 \pm 0.63$ ng m$^{-3}$, $0.89 \pm 0.89$ ng m$^{-3}$ and $1.02 \pm 0.69$ ng m$^{-3}$ at 8 m, 120 m and 260 m, respectively. The abundance of isoprene SOA tracers increased with height, contrasting with the vertical distributions of those BSOA from monoterpenes and sesquiterpenes.

The time series and relative contributions of the three kinds of BSOA tracers at the three layers are shown in Figure 2 and

Figure S6. Their vertical patterns in each day are also shown in Figure 3 and Figure 4. Generally, isoprene SOA tracers increased with height, while the other two kinds of SOA tracers varied slightly with height. The contribution of total isoprene SOA tracers to total BSOA tracers varied from 63% to 77% from 8 m to 260 m, that of monoterpene SOA tracers





from 33% to 21% and that of sesquiterpene SOA tracer from 4% to 2% (Figure 2d). As a result, the compositions of BSOA tracers showed increasing fractions of isoprene SOA tracers with height (Figure S7), which indicated that isoprene oxidized products were more important contributors to SOA in urban Beijing over the late summer than other BVOCs products. Such a pattern may also indicate that regional transport largely contributed to SOA from isoprene, while monoterpene- and

sesquiterpene-derived SOA were likely more influenced by local emissions. Our results are in agreement with the field observations over the United States and the modelled vertical distributions of isoprene-derived SOA, that is, high loadings of SOA from isoprene oxidation occurred above the surface layer (Zhang et al., 2007; Goldstein et al., 2009). In particular, each kind of BSOA tracers displayed different temporal and vertical distributions (Figure S6 and Figure 4). They are potentially influenced by multi-factors, such as oxidation processes (Claeys et al., 2004; Szmigielski et al., 2007) and emissions (Wang

et al., 2008; Faiola et al., 2014) of BVOCs, regional transport (Du et al., 2017), the mixing of heights (Wang et al., 2018b) and meteorological conditions of atmosphere (Ding et al., 2011).

**3.1.2 Impacts of the emissions of BVOCs**

Vegetation species, plant growth stage and environmental conditions can impact the release of BVOCs (Benjamin et al., 1997; Wang et al., 2003), which contribute to the complex vertical profiles of BSOA tracers. Northwest China is mainly

grasslands or barren lands, while other areas of China, especially the south of China, are rich in terrestrial plants (Ran et al., 2012). The emission inventory showed that in summer a large amount of BVOCs were mainly emitted from northeast, north, southeast regions with only a small amount from southwest China (Yan et al., 2005). Isoprene is one of the most abundant non-methane VOCs, mostly emitted by broadleaf plants (deciduous or evergreen trees) and marine phytoplankton (Sharkey et al., 2008; Stone et al., 2012). Considering air mass back trajectories which showed about 70% of the air masses

originating from south or northeast regions of Beijing (Figure S1), the aloft increased isoprene oxidation products was potentially influenced by the regional scale emissions of BVOCs from these regions. Monoterpenes are mainly emitted by needle leaf trees (e.g. coniferous plants), and the emissions from soil and litter in local places may be larger than those from vegetation (Faiola et al., 2014). Local emissions potentially resulted in the high concentrations of monoterpene SOA tracers (e.g. primary oxidized products: PNA and PA) at the ground surface layer. However, the average concentrations of

monoterpenes SOA tracers at three heights were not significantly different, suggesting the influence of mixing processes. Sesquiterpenes are mainly emitted by crops and herbs. The higher abundance of β-caryophyllinic acid at 8 m than at the other two layers (Figure 4m) is thus mostly associated with the local emission of sesquiterpenes.

The vegetation distribution is just one possible factor influencing on the vertical profiles of BSOA tracers. Terrestrial plantation can emit a broad spectrum of BVOCs, and ambient temperature, illumination, soil moisture and pollution situation

also affect their formation processes and concentrations in the atmosphere. All these factors with the addition of oxidation processes (such as vapor pressure of oxidation products, reaction rates and lifetime of BVOCs) and meteorological conditions simultaneously control the loading of BSOA (Tarvainen et al., 2005; Jaoui et al., 2007) and cause their complex vertical profiles in the UBL.



### 3.1.2 Vertical properties in the photo-oxidation of BSOA tracers

Isoprene SOA tracers are the photo-oxidation products of isoprene with atmospheric oxidants (e.g. OH, $O_3$ and $NO_x$). The reactions mainly include two mechanisms: formation of isoprene epoxydiols (IEPOX) intermediates at a low level of $NO_x$ and formation of methacryloyl peroxynitrate (MPAN, e.g. methacrolein, methyl vinyl ketone and methyl butanediols) at a high level of $NO_x$. These processes are influenced by many factors, such as atmospheric conditions (humidity, temperature and solar radiation) and the acidity of aerosols (Claeys et al., 2004; Surratt et al., 2007; Kleindienst et al., 2009; Surratt et al., 2010; Nguyen et al., 2015). Specifically, 2-MGA is mainly formed under a high $NO_x$ level and 2-MTs (the sum of 2-MET and 2-MT) are formed under a low $NO_x$ level. The ratio of 2-MTs / 2-MGA can reflect the impacts of $NO_x$ level on isoprene oxidation processes (Surratt et al., 2010). In this study, the average ratio of 2-MTs / 2-MGA was $5.20 \pm 2.24$ at 8 m, higher than the values at 120 m ($3.80 \pm 1.95$) and 260 m ($3.15 \pm 1.83$) (Figure 5a). The aloft lower ratio suggested aerosols transported from other polluted regions with higher $NO_x$ levels contributed to isoprene oxidation products in the upper layer aerosols of Beijing. The impacts of other factors (e.g. relative humidity, temperature and oxidizing capacity) on the heterogeneous oxidations of isoprene cannot be ignored (Wang et al., 2018a). The higher values of 2-MTs / 2-MGA in this study than in a previous study in Beijing at the ground level (average 1.7) (Ding et al., 2014) suggests the efficient decreasing of $NO_x$ level in the atmosphere under the strictly implementation of emission controls. In addition, the average values of another ratio, 2-MET / 2-MT, were ~ 2.5 at the three layers (Figure 5b), higher than New York and Mt. Tai (Xia and Hopke, 2006; Fu et al., 2010), implying their multiphase oxidation processes in the atmosphere .

Positive correlations of 2-MTs and $C_5$-alkane triols at the three heights (R > 0.5, Figure S8) indicated their similar formation pathways in the atmosphere (Surratt et al., 2006). The average ratios of 2-MTs / $C_5$-alkene triols were $0.97 \pm 1.17$, $1.33 \pm 1.24$ and $3.97 \pm 3.08$ at 8 m, 120 m and 260 m, respectively (Figure 5c). The increase with height of 2-MTs fractions in contrast to a decrease with height of $C_5$-alkene triols (Figure S7) indicated that isoprene-derived SOA was likely more aged at 260 m than at the ground surface layer because the formation of 2-MTs have been hypothesized to be subsequent products of $C_5$-alkene triols oxidation (Wang et al., 2005). The aloft increasing concentrations of 2-MTs and 2-MGA with small vitiations of $C_5$-alkene triols at three heights (Figure 4) indicated that the SOA at 260 m contributed more loading of 2-MTs and 2-MGA other than $C_5$-alkene triols in atmosphere aerosols. It also implied that photo-oxidation of isoprene was likely stronger at high height than at low height and-/or that these vertical patterns were related to aged aerosols transported at a regional scale.

Eight monoterpene SOA tracers were identified with pinonic acid (PNA), pinic acid (PA), MBTCA being the dominant compounds (Table S1). The different temporal and vertical patterns of these tracers are displayed in Figure S6 and Figure 4. MBTCA can be produced by further oxidations of PNA and PA by OH radical (Szmigielski et al., 2007; Ding et al., 2016). Thus, the ratio of MBTCA / (PNA+PA) can represent the ageing extent of monoterpene SOA. The ratio of MBTCA / (PNA+PA) at 8 m ($0.24 \pm 0.10$) was lower than those at 120 m ($0.84 \pm 0.44$) and 280 m ($1.49 \pm 0.77$) (Figure 5d) with increasing MBTCA fractions and decreasing PNA fractions (Figure S7) at upper layers, indicating that SOA from



monoterpenes was much fresher at the surface than the upper layers. The values of MBTCA / (PNA+PA) differed significantly with height (Table S2). These results suggested that the lower height (8 m) was more relevant to local fresh aerosols whereas the higher layer (260 m) was more subject to regional aged aerosols, and the middle layer (120) was likely the mixed influence of local and regional aerosols. This conclusion was also supported by the more significant correlation

between PNA and MBTCA at 8 m (r =0.7) than those at 120 m (-0.06) and 260 m (0.27) (Figure S8).

β-Caryophyllinic acid is produced by the oxidation of β-caryophyllene emitted from trees and plants (Jaoui et al., 2007). The average concentration of β-caryophyllinic acid decreased and then increased slightly with height (Figure 4), being related to relatively high ambient temperature (Duhl et al., 2008) or β-caryophyllene released from the soil or litter around the ground surface (Zhu et al., 2016). It is noteworthy that the correlations (r) of β-caryophyllinic acid with other SOA tracers (poly

acids, aromatic acids, 2-MGA, $C_5$-alkene triols and 3-hydroxyglutaric acid) were stronger at 120 m and 260 m than those at 8 m (Figure S7), implying that these tracers had the same origins and were potentially associated with regional transport of aerosols at upper layers.

### 3.1.3 Vertical profiles of BSOA tracers during pollution events

Generally, the total concentrations of BSOA tracers increased with height in the E1 (August 17th and 19th) and the E2

(August 29th), and complex vertical distributions were recorded during other pollution events. The lower concentration of BSOA tracers (13.2 ng m$^{-3}$) at 120 m on August 18th (E1) than average values (27.1 ng m$^{-3}$) during the whole sampling period was likely related to removal by a short-lived rain event. The air masses during the pollution episodes were mostly from the south of Beijing (Figure S2), which contributed to the formation of air pollution in the city (Zhao et al., 2017).

High abundance and increasing fractions of isoprene SOA tracers with height were recorded on August 17th and 19th of E1

and E2 (Figure 6), likely associated with the regional transport from southern areas of Beijing (Figure S2 and Figure S3). The aloft lower abundance of isoprene oxidation products than at the surface layer on August 16th (E1) was likely influenced by the air masses from the northwest. The same difference on September 7th to 8th (E3) was likely influenced by the air masses from the northeast. In addition, lower values of 2-MTs / $C_5$-alkene triols at 260 m were recorded during the pollution episodes when compared with other periods (Figure 5c), suggesting that air pollution can be adverse to the further photo-

oxidation of BSOA. Monoterpene SOA tracers during the pollution events showed vertical patterns similar to the average values, that is, the higher concentrations and fractions were recorded at the ground surface layer than at the upper layers due to local emissions. However, their concentrations increased with height on August 19th (E1) (Figure 3b), likely influenced by regional transport. Sesquiterpene SOA tracer showed unusually vertical distribution patterns during the episodes, that is, higher concentrations were recorded at the upper layers than the at the ground surface layer (Figure 3c), which was also

associated with the regional transport.

The vertical patterns of BSOA tracers during the pollution events highlighted the significant roles of air masses origins, regional transport, local emissions and oxidation processes on urban aerosols of Beijing. More field measurements are needed to address the interactions between SOA formation and the urban boundary layer.



### 3.2 Vertical profiles of DHOPA

DHOPA is an anthropogenic secondary organic compound, which is often used as a tracer for toluene (aromatic hydrocarbon) and which can only be detected in particle phase (Kleindienst et al., 2007; Al-Naiema and Stone, 2017). DHOPA concentrations were $0.90 \pm 0.53$ ng m$^{-3}$, $1.50 \pm 1.09$ ng m$^{-3}$ and $2.03 \pm 1.69$ ng m$^{-3}$ at 8 m, 120 m and 260 m, respectively. The average concentrations at 8 m and 260 m differed significantly. The increased concentrations of DHOPA with height contributed to regional transport. This vertical pattern was more obvious during pollution episodes, except for August 16th and September 7th when air masses were from the northwest and northeast (Figure S2). Thus, the increasing abundance of DHOPA at the upper layers during the episodes was most likely related to the pollutants from the southern region of Beijing. In addition, DHOPA correlated well ($r > 0.7$) with aromatic acids and polyacids at the three heights, suggesting that they had similar origins such as anthropogenic aromatic VOCs (Al-Naiema and Stone, 2017; Ding et al., 2017). DHOPA also showed good correlations ($r > 0.5$) with 2-MGA, C$_5$-alkene triols, 3-HGA and β-caryophyllene acid (Figure S7), suggesting interactions between anthropogenic and biogenic SOA. For instance, it has been reported that the existence of aromatic compounds (PAHs) can lead to high loading of α-pinene-derived SOA (Shilling et al., 2012; Zelenyuk et al., 2017). In addition, potentially similar sources, such as traffic transport which can simultaneously release isoprene and toluene (Borbon et al., 2001), can also result in positive correlations between DHOPA and some BSOA tracers.

### 3.3 SOC estimation by the tracer-based method

The tracer-based method is used to better estimate and investigate the contributions of different sources to SOC along vertical gradients. The fraction factors for SOC from isoprene, monoterpene and sesquiterpene (Iso_SOC, Mon_SOC and Sesq_SOC) are set as $0.155 \pm 0.039$, $0.231 \pm 0.111$ and $0.0230 \pm 0.0046$, respectively, and those for toluene SOC (DHOPA as tracer) and naphthalene SOC (phthalic acid as a surrogate) are $0.0079 \pm 0.0026$ and $0.0199$, respectively (Kleindienst et al., 2007; Kleindienst et al., 2010; Kleindienst et al., 2012). It should be noted that estimations of fraction factors in chamber processes deviated from real atmospheric environment (Lewandowski et al., 2013; Ding et al., 2014; Al-Naiema and Stone, 2017). Quantitative uncertainties, system errors, volatility of BSOA tracers and other factors could also increase the challenge in getting more accurate estimation of SOC. These uncertainties imply that more indoor experiments and field investigations are needed to integrate accurately the fractions of VOCs transformed to SOC.

Temporal variations in the estimated SOC and their percentages in OC at the three heights are shown in Figure 7 and Table S3. The sum of these reconstructed SOC were $341 \pm 150$ ngC m$^{-3}$, $444 \pm 283$ ngC m$^{-3}$ and $673 \pm 385$ ngC m$^{-3}$ at 8 m, 120 m and 260 m, respectively, with their average percentages in OC being $8.05 \pm 3.17\%$ (8 m), $8.60 \pm 3.66\%$ (120 m) and $13.4 \pm 4.81\%$ (260 m). Toluene SOC was the dominant contributor to SOC (32%, 41% and 35 % at 8 m, 120 m and 260 m, respectively), followed by naphthalene SOC and BSOC (Iso_SOC, Mon_SOC and Sesq_SOC). The sum of ASOC (toluene and naphthalene SOC) contributed more than 50% of these SOC at three heights, and their concentrations and fractions increased with the height (Figure 7c), suggesting a significant impact of anthropogenic sources from regional transport on



urban aerosols of Beijing. The average concentrations of BSOC ranged from 157 to 272 ngC m⁻³ and accounted for 3.80 ± 1.46% (8 m), 3.09 ± 0.97% (120 m) and 5.63 ± 2.32% (260 m) in OC (Table S3).

BSOC showed different fractions at the three heights. Iso_SOC fractions at the upper layers were higher than at the ground surface, while Mon_SOC and Sesq_SOC fractions at the ground surface were the highest (Figure S8). These features

illustrated the large contribution of regional transport to isoprene derived SOC above the surface layer, while monoterpenes and sesquiterpene were likely influenced by local emissions. Consequently, the fractions of toluene SOC and Iso_SOC increased with height, Mon_SOC and Sesq_SOC fractions decreased with height and naphthalene SOC fractions were similar at the three heights, suggesting that regional transport are rich in toluene SOC and Iso_SOC. In addition to the influence of local emission and regional transport, meteorological conditions, atmosphere turbulences and UBL movement

also cannot be ignored.

### 3.4 Impacts of emission controls on SOC loadings

The average concentrations of SOC before, during and after the Parade (Defore-, During- and After-Parade) are shown in Figure 8, with lower abundance during the Parade period. The sum of SOC during the Parade decreased by about 12% and 10% compared with the Before-P and After-P at 8 m, 35% and 16% at 120 m and 31% and 4% at 260 m, respectively. The

SOC at the upper layers decreased more than at the ground surface layer, suggesting the efficient mitigation of SOC on a regional scale. During the reduction period, ASOC fraction at 8 m obviously decreased by 12% and Iso_SOC fraction increased by 1–7%; ASOC fractions decreased by 3–5% and Iso_SOC fraction increased by 6–7% at 120 m; ASOC fractions decreased by 10% and Iso_SOC largely increased by 10–13% at 260 m. The decreased contributions of ASOC during the control period indicated the emission controls were effective in mitigating anthropogenic sources, with the controlling on

toluene SOC being especially effective. However, emissions mitigation was not so efficient to control BSOC levels, especially Iso_SOC, implying that SOA from isoprene oxidation was potentially a more stable contributor than other VOCs in the urban Beijing during late summer. Consequently, these results indicate that regional emission controls changed the aerosol SOC composition. Moreover, meteorological conditions and other factors (e.g., atmospheric oxidation state) could also impact the variations in SOC during different sampling periods, such as the wind shift prior to and after the Parade and

the complex vertical distributions of particulate nitrate (Zhao et al., 2017; Wang et al., 2018a; Wu et al., 2019).

## 4 Conclusions

The vertical profiles of biogenic and anthropogenic SOA tracers in urban Beijing in the late summer were investigated in this study. The sum of BSOA tracers were 31.5 ± 16.8 ng m⁻³ (8 m), 36.4 ± 26.1 ng m⁻³ (120 m) and 50.2 ± 27.0 ng m⁻³ (260 m). BSOA from isoprene was the dominant compound, followed by monoterpenes and sesquiterpene. The fractions of isoprene

SOA tracers showed an aloft increasing vertical pattern, whereas monoterpene and sesquiterpene SOA tracers showed opposite variations. These vertical characteristics of BSOA tracers were influenced by multiple factors, such as their photo-



oxidation processes, local emissions and regional transport of their precursors, as well as mixed influences. We concluded that isoprene oxidation products were more influenced by air masses from regional transport, and monoterpene oxidation products were more influenced by local emission sources. The specific vertical distributions of BSOA tracers during pollution episodes suggested a significant contribution of regional transport of aerosols from the southern regions of Beijing.

The average concentrations of the toluene tracer (DHOPA) were $0.90 \pm 0.53$ ng m$^{-3}$ (8 m), $1.50 \pm 1.09$ ng m$^{-3}$ (120 m) and $2.03 \pm 1.69$ ng m$^{-3}$ (260 m). DHOPA showed an aloft increasing patterns with larger variations during the episodes, also suggesting the regional transport of pollutants from the southern regions.

Estimated by the tracer-based method, the sum of SOC was $341 \pm 150$ ngC m$^{-3}$ (8 m), $444 \pm 283$ ngC m$^{-3}$ (120 m) and $673 \pm 385$ ngC m$^{-3}$ (260 m), with toluene SOC being the dominant compound, followed by naphthalene SOC, Iso_SOC and other SOC. The aloft increasing SOC suggested a contribution of regional transport. The increase in toluene SOC and Iso_SOC fraction with the height indicated that the air masses subject to regional transport were potentially rich in toluene- and isoprene-derived SOC. The implementation of joint regional prevention and control by the government can significantly reduce the amount of SOC, however, they are likely more efficient on reducing toluene SOC, but not isoprene-derived SOC. Our study demonstrates the variability of SOA within the urban boundary layer and highlights that vertical profiles of SOA are critical to improving the simulation of SOA in chemical transport models.

*Data availability.* The atmospheric particulate matter data used for analysis are available in the Supplementary Material, and the data are also available upon request from the corresponding author.

*Competing interests.* The authors declare that they have no conflict of interest.

*Acknowledgements.* This work was supported by the National Key R&D Program of China (Grant No. 2017YFC0212700), the National Natural Science Foundation of China (Grant No. 41625014, 41475117, 41571130024) and China Postdoctoral Science Foundation (Grant No. 390/0401130003). The vertical meteorological data was obtained from the Institute of Atmospheric Physics (IAP), Chinese Academy of Sciences (CAS). The tower samples were collected with the help of staffs of IAP. Detailed tables and figures about the data in this manuscript are present in the supporting information. The language of this manuscript has been edited by International Science Editing (http://www.internationalscienceediting.com).

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



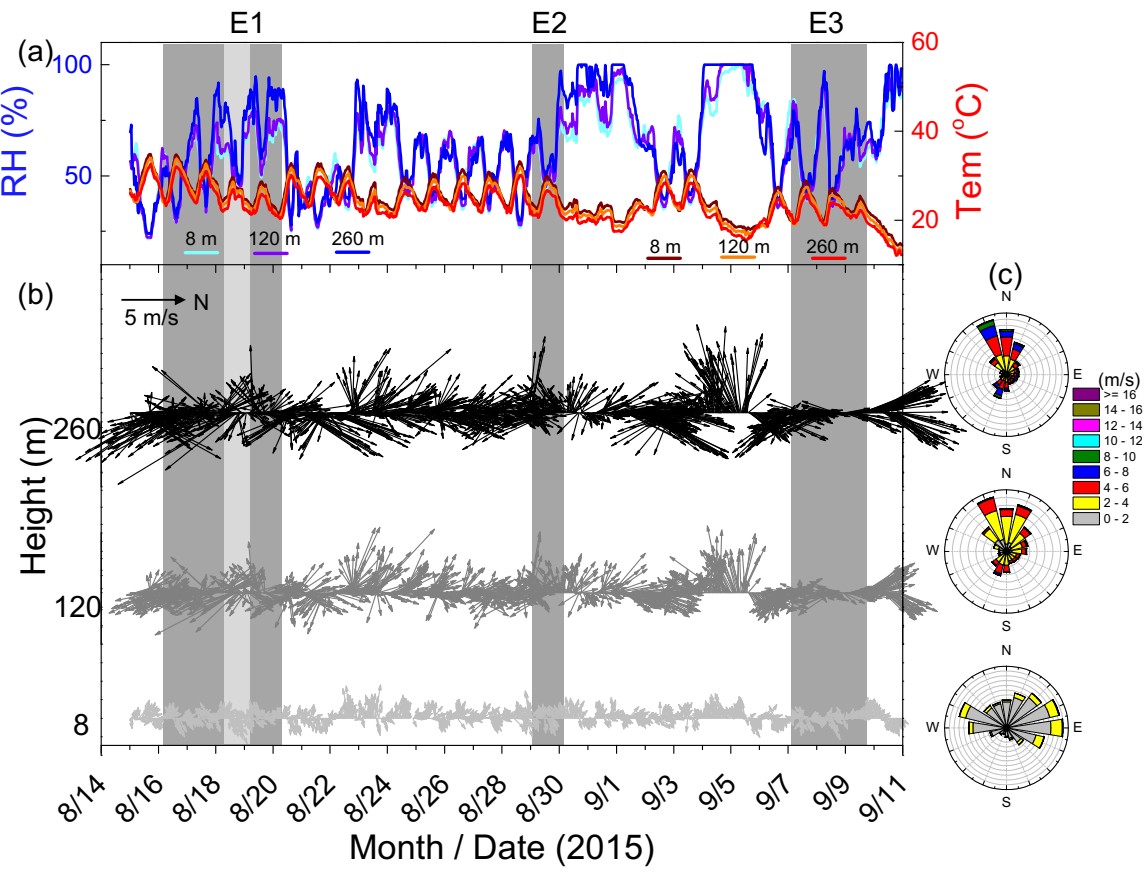

**Figure 1. Temporal series of vertical meteorological parameters including (a) relative humidity (RH) and temperature (Tem), (b) wind direction (WD) and wind speed (WS), and (c) wind roses. These data were measured at 8 m (a two-story building), 120 m and 260 m (the 325-m meteorological tower), respectively. Three pollution events (including E1 to E3) are indicated by grey shading (E1: August 16th to 19th, E2: August 29th and E3: September 7th to 8th, respectively). The light grey shading during E1 is a short rain event which reduced the loading of OA in aerosols. N represent the northern wind. Obvious meteorological conditions were found during the sampling period.**



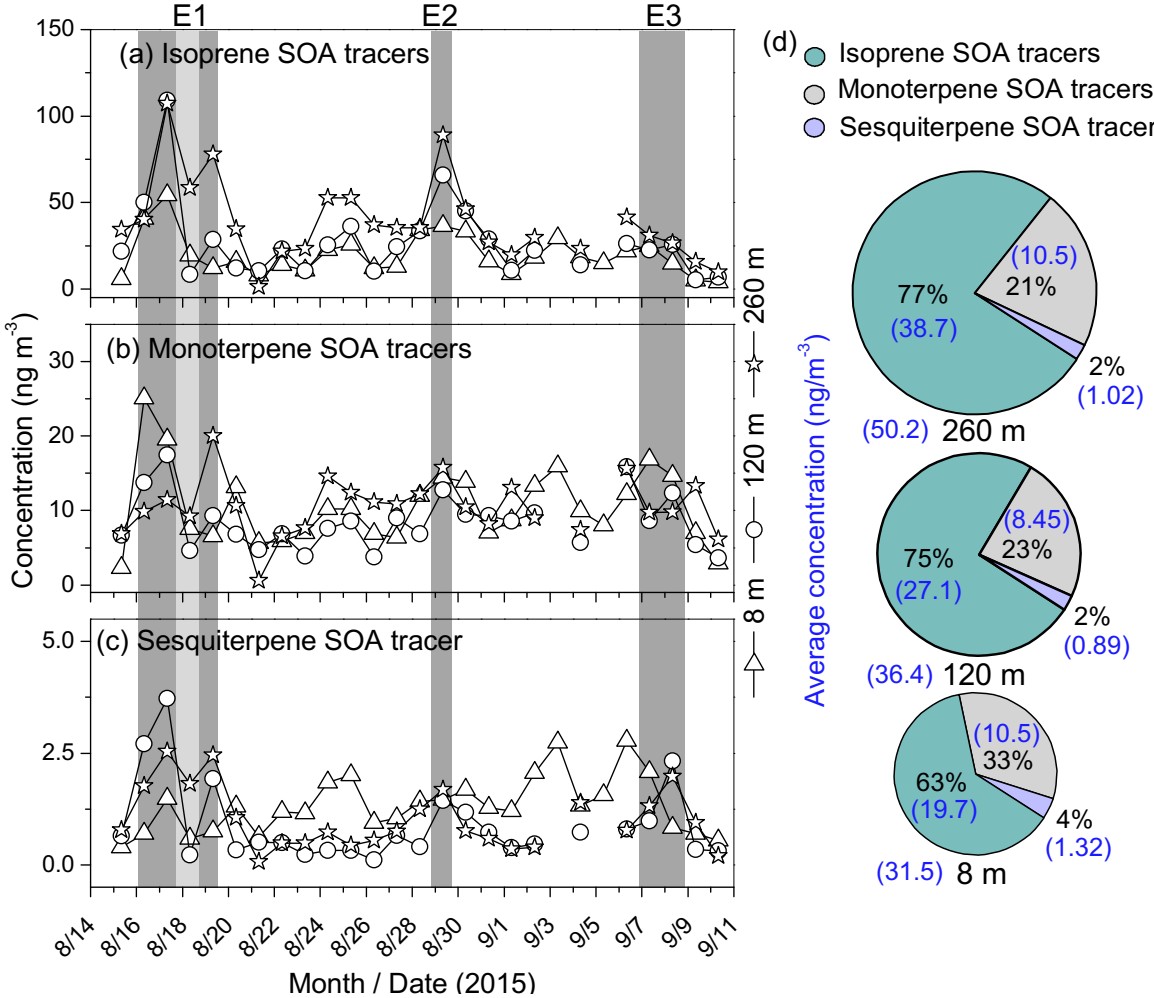

**Figure 2. Vertical and temporal variations in (a) Isoprene SOA tracers, (b) Monoterpene SOA tracers, (c) Sesquiterpene tracer. These data were measured at 8 m (marked as triangle), 120 m (marked as circle) and 260 m (marked as star) in PM$_{2.5}$ of Beijing. (d) Mass fractions of BSOA from isoprene, monoterpene and sesquiterpene. These figures mainly display the vertical profiles of BSOA tracers.**

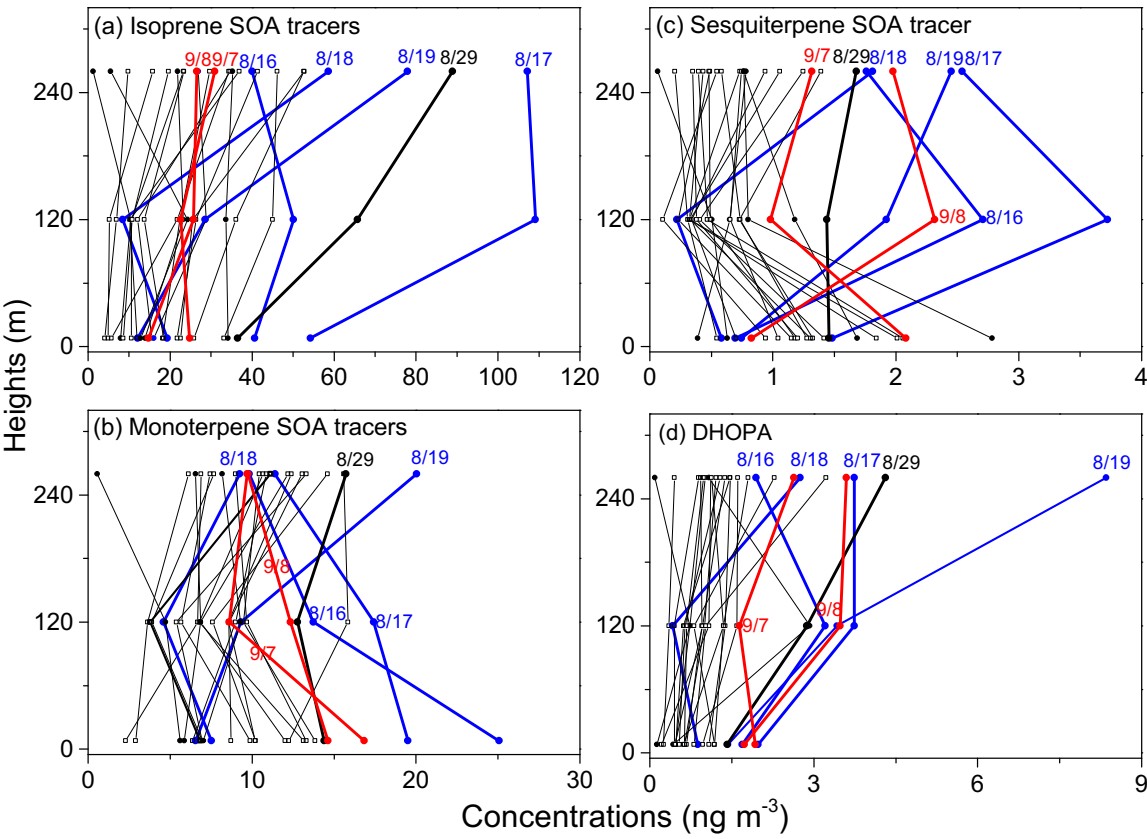

**Figure 3. Vertical distributions of BSOA tracers: (a) Isoprene SOA tracers, (b) Monoterpene SOA tracers and (c) Sesquiterpene SOA tracer, and ASOA tracers: (d) DHOP in each PM$_{2.5}$ sample during the sampling period. The samples collected during E1, E2 and E3 periods are marked with blue, black and red lines, respectively. The sampling date during the pollution events also marked near the vertical lines with the same color. These results imply the common influence of multi-factors on the complex vertical distributions of SOA tracers.**





**Figure 4. Vertical gradients of SOA tracers: (a–d) Isoprene SOA tracers, (e–i) Monoterpene SOA tracers, (m) β-Caryophyllinic acid, (n) Dihydroxy-4-oxopentanoic acid (DHOPA) and (o) Phthalic acid in PM$_{2.5}$ of Beijing. Figure (a-m) and (n-o) are tracers of**
5 **BSOA and ASOA, respectively. Vertical distributions of these tracers suggest their various fates in the atmosphere.**





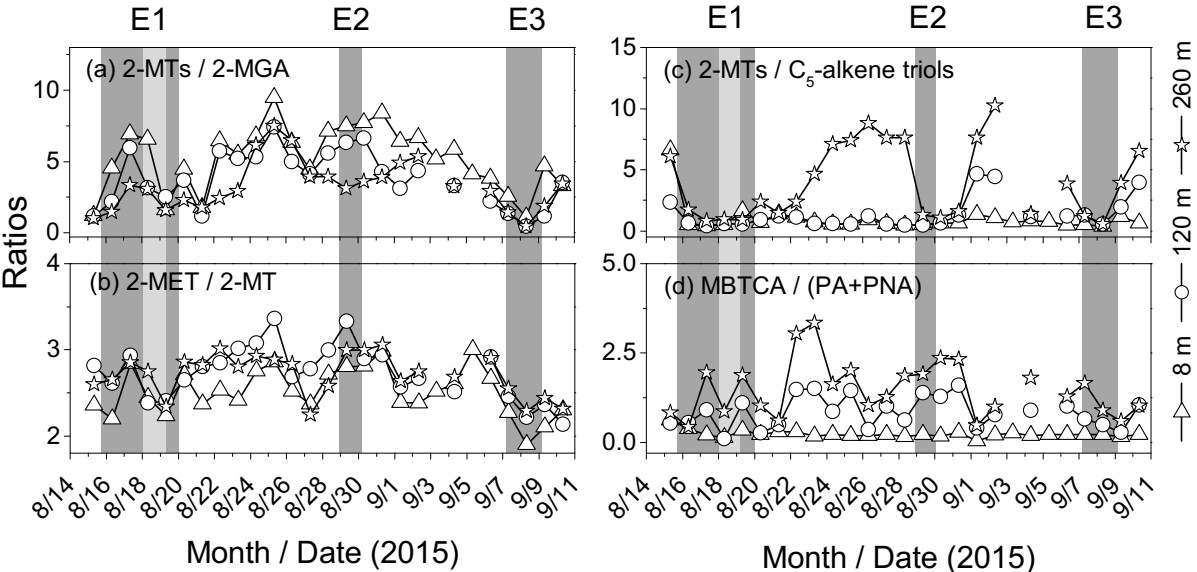

**Figure 5. Vertical and temporal variations in the concentration ratios of biogenic SOA tracers: (a) 2-MTs / 2-MGA ( the influence of NO$_x$ level, high NO$_x$ is favoured for 2-MGA while low NO$_x$ is favoured for 2-MTs), (b) 2-MTs / C$_5$-alkene triols (the aging state of isoprene products, 2-MTs are often suggested to secondary oxidation products of C$_5$-alkene triols), (c) 2-MET / 2-MT (the reaction route of isoprene) and (d) MBTCA / (PA+PNA) (the aging state of monoterpene products, PA+PNA are often suggested to secondary oxidation products of MBTCA) in PM$_{2.5}$ of Beijing.**





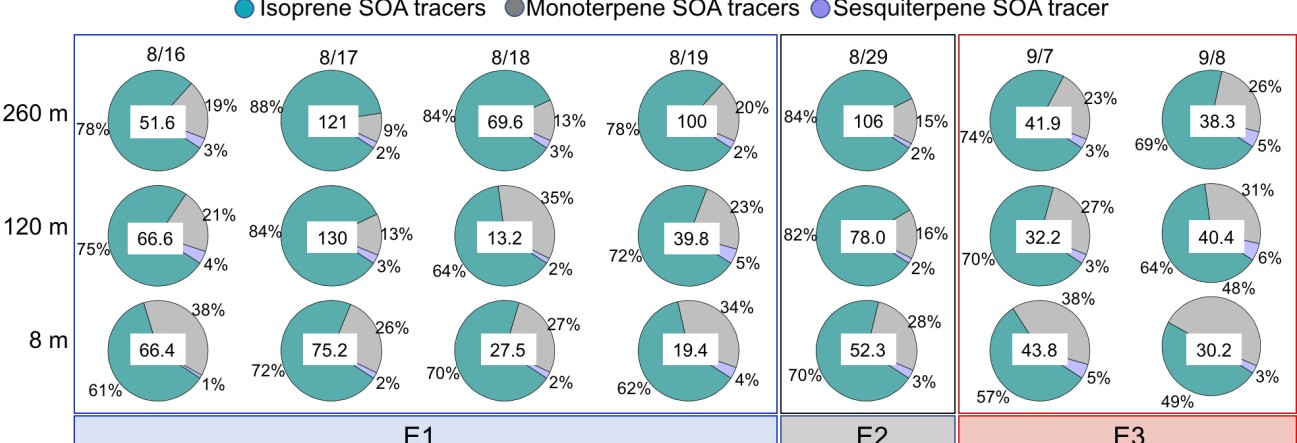

**Figure 6. Relative contributions of three kinds of BSOA tracers in samples collected during the pollution events in PM$_{2.5}$ of Beijing. The sum concentrations of BSOA tracers (ng m$^{-3}$) are showed in the centre of each pie. The mass fractions of SOA from isoprene increased versus from monoterpene decreased with the increasing height in the urban boundary layer.**

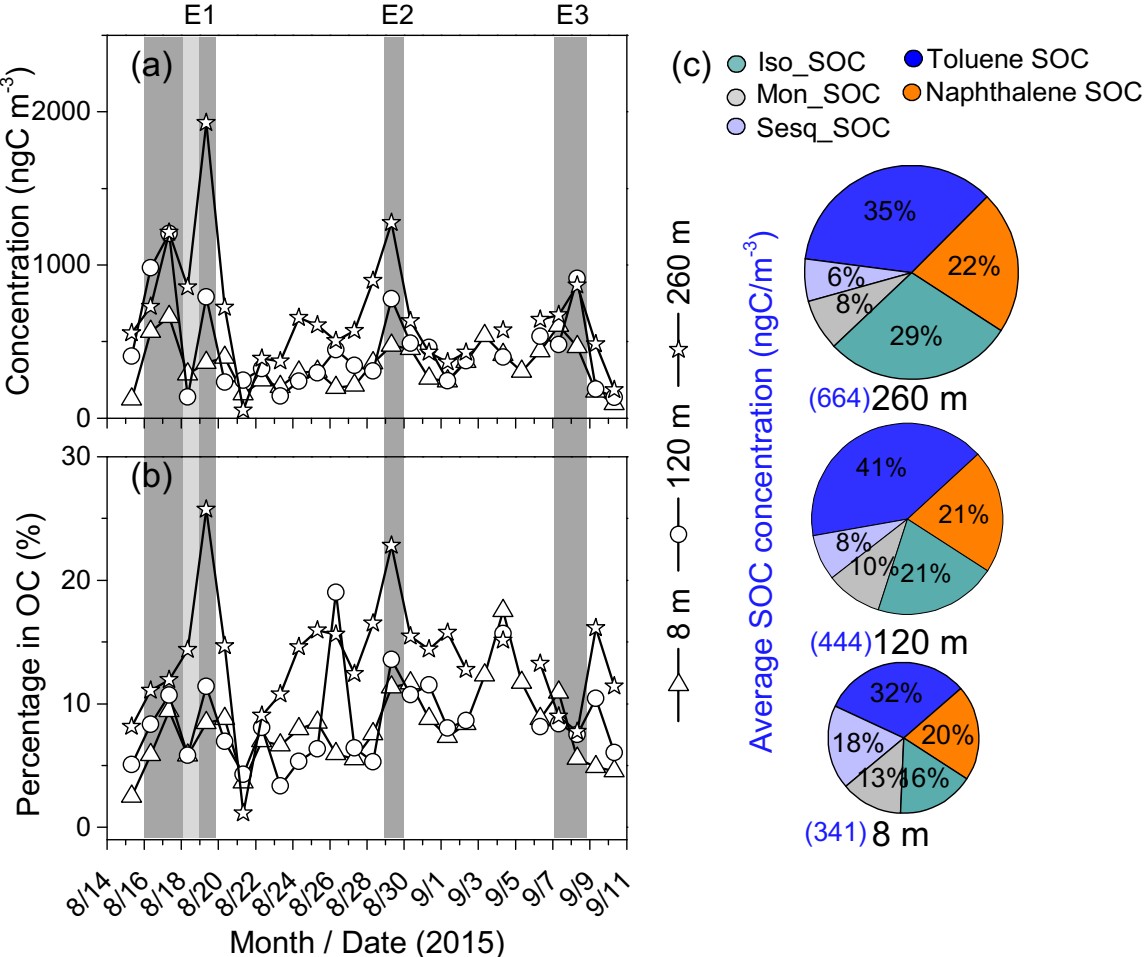

**Figure 7. Temporal variations in and relative contributions of estimated SOC based on tracer method in PM₂.₅ at 8 m, 120 m and 260 m in urban Beijing. (a) Vertical profiles of the sum concentrations of estimated SOC, (b) vertical profiles of estimated SOA percentage in OC and (c) vertical profiles of mass fractions of estimated SOC. Iso_SOC, Mon_SOC and Sesq_SOC represent BSOC estimated from isoprene, monoterpene and sesquiterpene, respectively. Toluene SOC and naphthalene SOC represent ASOC that were estimated from DHOPA and phthalic acid, respectively.**




**Figure 8. Temporal variations in and mass fractions of estimated SOC during three periods (Before-P means before Parade: August 15th to 19th; During-P means during Parade (light blue shading): August 20th to September 3rd and After-P means after Parade: September 4th to 10th.) at 8 m, 120 m and 260 m in PM2.5 of Beijing. The values in the centre of the pies represent the average concentrations of estimated SOC (ngC m-3) and the sizes of the pie are related to the concentrations.**