# Peer review of "Measurement report: Vertical distribution of biogenic and anthropogenic secondary organic aerosols in the urban boundary layer over Beijing during late summer"

_Atmospheric Chemistry and Physics, 2021_

## Author Comment (AC1)

**Responses to Referee 1**

We are thankful to the reviewer for the thoughtful comments and suggestions. We have revised the manuscript accordingly. The listed below are our point-by-point responses in blue to the comments. The modified parts in the revised manuscript are highlighted in yellow.

**RC1:** Review of "Measurement report: Vertical distribution of biogenic and anthropogenic secondary organic aerosols in the urban boundary layer over Beijing during late summer" by Ren et al.

The manuscript describes observational results of SOA tracers from a tall tower located in Beijing at the end of summer 2015 for about 5 weeks, which encompassed a period of tighter emission control in the end of august. Daily $PM_{2.5}$ sampling was conducted at three different heights, allowing in turn to study the vertical profiles of biogenic and anthropogenic tracers. This is a quite interesting dataset, particularly showing how different heights ranging from 8m up to 260m at an urban site can lead to quite distinctive conclusions regarding the contribution of SOA precursors. I do identify though some major issues that need to be addressed prior acceptance.

We thank the reviewer's encouraging comments. All comments and suggestions have been considered carefully and well addressed below.

*General comments:*

**#1**– Interpreting changes in SOA (or their tracers), is highly complex because it depends on meteorology (particularly for BSOA), air mass transport, pre-existing aerosol population and so on. The manuscript generally assumes that if concentration at 260m is higher than at 8m, then it's regional, otherwise local, but this is oversimplified (a local VOC source could also produce maximum SOA at 260m high, depending on vertical mixing, oxidation time, etc.). Here are some suggestions to help data interpretation: i) Provide a significantly thorough site description. As most of those tracers can be formed within hours (or less), there is a high potential of a source being local. For example, what is the vegetation cover and its type surrounding the site? ii) prior performing back-trajectories, analyze polar plots of tracer concentration vs wind speed/direction to help identify local vs regional formation; iii) add information of meteorology (particularly including solar radiation) on interpreting SOA tracers temporal variability, which is particularly important on assessing the effects of strict emission controls, but also the pollution events. iv) add any possible ancillary measurements (CO, NOx, $O_3$, VOCs) that could help better interpret the observations. For example, if CO and DHOPA is higher at 260m than 8m, than its regional contribution is obvious. Eventually EC could also be used as normalizing parameter; v) add information on $PM_{2.5}$ levels, and if possible, its

composition, especially during pollution or parade period to link SOA tracers with PM composition.

Thanks for the reviewer's valuable suggestions, which greatly improve the quality of our manuscript.

According to the nice suggestions of the reviewer:

i) We have added the description of the sampling site and vegetation types. Please see Section 2.1 in the revised manuscript (on page 3 lines 23-28).

"The sampling site is at the Institute of Atmospheric Physics (IAP), Chinese Academy of Sciences (39°58.53′N, 116°22.69′E), which is in an urban site (between 3- and 4-ring) of Beijing and surrounded by street road (~50 m), highway (~300 m), a public park (~500 m to the southwest), restaurants (~100 m), residential housing and a gas station (~200 m). The predominant vegetation types surrounding the sampling site are deciduous broadleaf vegetation (acacia and juglandaceae), shrub, and lawn. The vegetation cover of the public park is more than 50%. The predominant vegetation is also deciduous broadleaf."

ii) Our samples are collected on a daily basis, and the resolution of the concentrations of SOA tracers is low. If we use the average daily values of wind speed/direction vs the daily concentrations of SOA tracers to do polar plots, it should introduce large errors. So, we do not analyze polar plots of tracer concentration vs wind speed/direction in our revised manuscript. We hope to get the reviewer's understanding.

iii) We have added the discussion of meteorology on the effects of air pollution and the emission control period. Please see the part of Section 3.1.4 and 3.4 (on pages 9 and 11). However, it is a pity that we do not obtain the solar radiation data to interpret SOA tracers.

iv) It is regretful that we did not obtain the vertical data of these ancillary parameters (CO, NOx, $O_3$, VOCs), but we got the ancillary parameters (including CO, $NO_2$, $SO_2$, $O_3$) at the ground surface about 3 kilometers away from our sampling site. We have also added the data of EC according to the reviewer's suggestion. These data are shown in Figure S4 and demonstrated in Section 2.5 (on page 5 lines 1-5).

[Figure]

Figure S4. Time variations in (a) OC, (b) WSOC, (c ) EC, (d) WSOC/OC, (e) EC/OC, and (f) to (i) are the levels of $PM_{2.5}$, CO, $NO_2$, $SO_2$ and $O_3$ from the monitor station of the Olympic center near the sampling site, respectively.

"The ground surface concentrations of $PM_{2.5}$, CO, $SO_2$, $NO_2$, and $O_3$ were obtained from the monitor station of the Olympic center (39.98°N, 116.40°E) about 3 kilometers away from our sampling site, which is available on the National urban air quality and real-time publishing platforms (http://106.37.208.233:20035/). The hourly levels of these parameters were shown in figure S4"

Despite these ancillary parameters can not be used to explain the vertical properties of SOA tracers, they can be used to explain the formation of haze in the summer of Beijing. These data are useful for improving the quality of our manuscript. We are very thankful for the nice suggestion of the reviewer.

v) It is a pity that we only obtained the concentrations of $PM_{2.5}$ at the ground surface about 3 kilometers away from our sampling site. These data are showed in Figure S4 and Section 2.5 in the revised manuscript. We did not obtain the composition of $PM_{2.5}$, we hope to get the reviewer's understanding.

**#2**– I suggest to change the order section 3 is presented. As it stands it starts highly descriptive and offers only generic interpretations (as P6L7-L12, for example) to explain the dataset. Then, some possible impacts of BVOCs (3.1.2) is given, and then finally the actual tracers are used to interpret the data, considering its oxidation steps and different branching, which is the main advantage of such methods compare to bulk analysis such as WSOC or AMS-like source-apportionment. I suggest beginning this section with a discussion on VOC sources, then, as the

tracer profiles are presented, interpret them using first and later stage oxidation products, as well as different branching's.

Thanks for the reviewer's valuable suggestions. The sources of VOCs are the important factor on the vertical profiles of SOA tracers. So, we changed the order of section 3.1 according to the suggestions of the reviewer. Section 3.1 was beginning with a discussion on VOCs emissions, then, the vertical profiles of BSOA tracers. Please see these changes in the revised manuscript.

**#3**– I invite the authors to give it a careful and complete read to ensure high quality text. I found several typos and reported on technical comments, but it's likely that I missed some.

We modified the typos and technical comments according to the reviewer's suggestions, and we also carefully revised our manuscript. We hope this revised manuscript could be a high-quality text.

**#4**– Lastly, I find that the number of references can be significantly reduced, by at least a factor 3. Reducing the number of references will improve the readability with a clearer information tracing. For broad claims such as P.2L.2, all those 8 references could be replaced by to the latest IPCC report, for example.

Thanks for the reviewer's suggestion. We have reduced approximately one-third of the references. We have also updated some references in our revised manuscript. Please see these changes in the revised manuscript.

*Minor comments:*

Could you please use colors instead of circle, stars and triangles for the three heights into all plots? It would significantly improve readability.

Thanks for the valuable suggestion of the reviewer. We have used colors instead of shapes for three heights into all plots to improve the readability of figures in our manuscript. Please see the revised manuscript.

**P3L19-22:** As curiosity, is there simulation results that could complement the results presented here?

As far as we have known, there are few simulation studies related to these results (Li et al., 2017; Miao et al., 2015; Zheng et al., 2015). The results presented here are mainly referenced in these studies on the vertical characteristics of aerosols in the field campaign in Beijing (Wang et al., 2018a; Wang et al., 2019; Zhao et al., 2017)

**P4L24**: I'm not an expert on this type of analysis, but I understand that recovery rate is an important part of the quantification process. Why recoveries were not used for correction here?

As mentioned in the reference (Fu et al., 2009), only several SOA standards are commercially available. The recoveries of such standards are generally higher than 80%. The quality and

quantity of other SOA tracers were obtained by comparison with those of literature data or by using the surrogates, so it is difficult to correct the recoveries with standards.

**P4L29-P5L5:** several minor issues and confusing sentences, please rewrite them in a clearer manner.

Thanks. We have changed these sentences to "Cluster analyses were applied to estimate the influence of air mass. As shown in Figure S1, seven clusters were determined. Air mass from south, southeast, and northeast of Beijing accounted for >70%. Especially, for pollution days, retroplumes of air masses were calculated by the FLEXPART (FLEXible PARTicle dispersion) model (Figure S2). Detailed information about the model was described in a previous study (Wei et al., 2018). The model was set with a height of 300 m (a.g.l.) and three-day backward trajectories." Please see it in the revised manuscript (on page 4 lines 26-30).

**P5L7:** please define which additional information.

We have changed this sentence to "Meteorological parameters (wind speed, wind direction, temperature, and relative humidity) at the sampling site during the observation period are shown in Figure 1. These meteorological parameters have been reported in the previous study (Zhao et al., 2017)" in the revised manuscript (on page 5 lines 7-9).

**P5L9:** How close where the buildings surrounding the sampling site?

The sampling site is in the urban area of Beijing, so many buildings are near the sampling site. Some high buildings are about half to several hundred meters away from the sampling site. We have added the sentence "Some high buildings are about half to several hundred meters away from the sampling site." in the revised manuscript (on page 5 lines 11-12).

**P5L12-14:** It's difficult to see from the plot, but it seems that at times (e.g. end of E1) there are at most 1-2 degrees difference between lowest and highest level, but >10% RH difference between 120m and 260m. The same is not observed during E3, for example. Why is that? It could be interesting to add solar radiation on this plot, for example.

We can see the average temperatures decreased about 1-2 degrees versus RH increased about 1%-5% between the lowest and highest level (Table S1). As mentioned by the reviewer, the variations of temperature and RH are obvious differences during E1 compared with E3. We think that the short rain event during August 18$^{th}$ can cause the decreasing of temperature and the wind shear can cause the differences of RH during E1. The southwest winds carry high RH and high pollutant air masses to the urban of Beijing (Wang et al., 2018b), which may be the reason for the vertical differences of RH during E1. It suggests that the formation of E1 is largely related to regional transport. The small vertical differences of RH during E3 suggest that complex pollution. These reasons are potentially the reason for the larger difference of SOA tracers during E1 than E3. We have added the discussion of meteorological conditions on air pollution, please see Section 3.1.4 in the revised manuscript.

We regret that we did not obtain the solar radiation data.

**P5L15:** Please rewrite.

Thanks. We have changed this sentence to "Three pollution episodes (marked as E1, E2 and E3) were recorded during the sampling period." (on page 5 line 16)

**P5L15:** How were defined the pollution episodes?

We defined the pollution episodes according to the previous study (Zhao et al., 2017) and air quality index (AQI) from the Chinese national environmental monitoring center. We have explained this in the revised manuscript. (on page 5 lines17-18)

**P5L20:** as it stands, it's difficult to compare OC and WSOC between heights and with variability (std, I assume?), perhaps target only a few values, for the rest it's listed on Table S1. WSOC and OC at three heights showed no significant differences. The comparison here just given a general summary.

**Fig. S1-S2:** Have you performed a polar plot analysis of SOA concentration considering wind direction and intensity? This would help identify the role of local vs regional sources before assuming all is long-range transport and could be explained by back-trajectories.

We are sorry that we did not analyze polar plots of the concentrations of SOA tracer. As mentioned above, the SOA concentrations are whole-day averaged and the resolution of SOA concentrations is not fitting for polar plots.

**Fig. S4:** What are the values showed to the right on the vertical profile? Average and std?

Thanks. The values at the right axis were average and std concentrations. Now, we have modified Figure S4 in the revised manuscript.

[Figure]

Figure S4. Time variations in (a) OC, (b) WSOC, (c ) EC, (d) WSOC/OC, (e) EC/OC, and (f) to (i) are the levels of $PM_{2.5}$, CO, $NO_2$, $SO_2$ and $O_3$ from the monitor station of the Olympic center near the sampling site, respectively.

**Table S1:** Is it for the whole period or just during the period impacted by restrictions linked to the parade. Please correct the caption if that's not the case.

We are sorry for this misunderstanding. The dataset in Tables S1 is for the whole sampling period. We have corrected the caption.

**Table S2:** This table is not very clear, with the a's, b's and b^b's. If the objective is identify statistically meaningful difference those can be indicated in bold, for example. Also, please rewrite the caption (perhaps "difference" was meant?).

According to the suggestion, we have revised Table S2 as below in the revised manuscript.

Table S2. Results of single factor analysis to test the significantly different of these average concentrations at tree heights.

| Component (ng m$^{-3}$)$^a$ | 8 m | 120 m | 260 m |
|---|---|---|---|
| Isoprene SOA tracers | 19.7±12.0 **b**$^b$ | 27.1±22.4 **b** | 38.7±24.1 **a** |
| Monoterpene SOA tracers | 10.5±5.18 **a** | 8.45±3.68 **a** | 10.5±3.86 **a** |
| β-Caryophyllinic acid | 1.32±0.63 **a** | 0.89±0.89 **b** | 1.02±0.69 **ab** |
| DHOPA | 0.90±0.53 **b** | 1.50±1.09 **ab** | 2.03±1.69 **a** |
| Phthalic | 2.66±1.27 **b** | 3.59±2.54 **a** | 5.17±2.89 **a** |
| WSOC | 2.03±0.99 **a** | 2.69±1.55 **a** | 2.73±1.31 **a** |
| OC | 4.37±1.69 **a** | 5.32±2.88 **a** | 5.03±2.28 **a** |
| WSOC / OC (%) | 46.9±11.9 **b** | 51.1±8.88 **a** | 54.0±5.63 **a** |
| 2-MTs / 2-MGA | 5.20 ± 2.24 **a** | 3.80 ± 1.95 **b** | 3.15 ± 1.83 **b** |
| 2-MET / 2-MT | 2.52±0.28 **b** | 2.73±0.31 **a** | 2.73±0.22 **a** |
| 2-MTs / C5-alkene triols | 0.97±1.17 **b** | 1.33±1.24 **b** | 3.97±3.08 **a** |
| MBTCA / (PAN+PN) | 0.24±0.10 **b** | 0.84±0.44 **b** | 1.49±0.77 **a** |

$^a$ The concentrations of these components are expressed as mean ± STD;

$^b$ Different lowercase letters in bold indicate significant differences at $P < 0.05$ of the mean concentrations of these compounds in aerosols collected at three heights. The same lowercase letters in bold indicate no significant differences.

**P6L7-11:** Globally I agree with the three points indicated by the authors, but I do not classify them equally to explain the differences on tracer levels among the three heights. I believe that it's a local vs regional impact (argument #2) that explains such variability. This is a strong result presented by the paper, raising a caveat on observations conducted at 8m (which is already quite high for typical urban sites, ranging usually to 3 or 4 meters) as representative of regional chemistry to be compared with meso-scale 3d models, for example.

Thanks for the reviewer's comments. We also think that local and regional transport are the

main reasons causing such variability of SOA tracers. Hence, we have rewritten these points. Please see "They are potentially influenced by multi-factors. The predominant reason is likely related to local emission and regional transport (Du et al., 2017). Secondly, the mixing of heights (Wang et al., 2018b) and meteorological conditions of the atmosphere (Ding et al., 2011) is potentially another important factor. Moreover, oxidation processes (Claeys et al., 2004; Szmigielski et al., 2007) and emissions (Faiola et al., 2014; Wang et al., 2008) of BVOCs can also cause this complex vertical profiles of SOA." in the revised manuscript (on page 7 lines 28-32).

8 m may be a little high for a typical local urban site, but it was usually thought to be representative of local sources in many previous studies (Du et al., 2021; Sun et al., 2015). We regret that we have no condition to do mesoscale 3d models to comparison with the results of 8 m for the restriction of conditions. We hope to get the understanding of the reviewer.

**Figure2:** As suggestion, the caption could be "SOA tracers of (a) isoprene, (b) monoterpenes and (c) sesquiterpenes. Measurement heights are 8m (triangles), 120m (circles) and 260m (star) in PM2.5. Relative mass fractions are shown in (d)." I remind also the authors that monoterpenes and sesquiterpenes make a group of several species (unlike isoprene, which is a single compound), so they should be referred in plural. I suggest modifying other captions as well to reduce repetitions and make easier to understand.

We have changed the caption of figure 2. Please see "Figure 2. Vertical and temporal variations in BSOA tracers from (a) isoprene, (b) monoterpenes, and (c) sesquiterpene. Measurement heights were at 8 m (solid circles), 120 m (grey circles), and 260 m (open circles). Relative mass fractions are shown in (d)." in the revised manuscript (on page 19 lines 3-5).

Because we only found one tracer of sesquiterpene oxidated products, the sesquiterpene is used in our revised manuscript.

We also have modified other captions in our revised manuscript.

**Figure3:** Figure difficult to read.

We have modified the caption of figure 3 to "Vertical profiles in the concentrations of SOA tracers from (a) isoprene, (b) monoterpenes (c) sesquiterpene and (d) DHOP in each day sample collected at three heights. The samples collected during E1, E2, and E3 periods are marked with blue, black, and red bold lines, respectively. The sampling date during the pollution days is also marked." in the revised manuscript (on page 20 lines 3-5).

**P7L2-3:** Be careful not to mix tracer concentration with SOA concentration (as later discussed in section 3.3).

We have deleted this sentence. We emphasized that the concentration of SOC in Section 3.3 (now is Section 3.4 in the revised manuscript) was estimated according to the tracer-based method. Now, it should do not mix with SOA tracer concentrations.

**Figure 7:** perhaps would be more interesting to compare sum of SOC to WSOC (as a proxy for total SOC), it would probably correspond to about 50% of total SOC.

The fraction factors of SOC using in the tracer-based method are obtained by comparing with OC. So, we only compare the sum of SOC to OC in our study.

**Section 3.1.3:** It could be interesting to calculate enrichment factors during the pollution events (perhaps normalized by deltaEC, or deltaCO, if available, from non-pollution periods).

Thanks for the reviewer's valuable suggestion. Considering that we didn't get the vertical data of CO, we did not discuss the enrichment factor of CO during the pollution events. However, we have added a brief discussion about the enrichment factor of EC during the pollution events. Please see "The concentrations of EC and the ratio of EC / OC (Figure S4) showed extremely low values and vertical varies during E2 when compared with other pollution events, suggesting that the formation of E2 is largely influenced by regional transport. In addition, the increasing levels of pollution parameters (such as $O_3$, $SO_2$, and $NO_2$) also contributed to the pollution episodes." in Section 3.1.4 in the revised manuscript (on page 9 lines 13-16).

**P10L10-L15:** Be mindful that correlation and causality are not the same thing. The fact that there is correlation between isoprene tracers and DHOPA, or that traffic can emit some VOC is not itself an indicative of biogenic-anthropogenic interaction.

We have rewritten these sentences. Please see "DHOPA also showed moderate correlations ($r >$ 0.5) with 2-MGA, $C_5$-alkene triols, 3-HGA, and β-caryophyllene acid (Figure S7). Previous studies have reported that urban pollution can enhance the formation of natural aerosols (Shrivastava et al., 2019); the existence of aromatic compounds can lead to high loading of α-pinene-derived SOA (Shilling et al., 2012; Zelenyuk et al., 2017); and traffic transport can simultaneously release isoprene and toluene (Borbon et al., 2001). These results suggest that anthropogenic sources can impact the formation of biogenic oxidation products." in the revised manuscript (on page 10 lines 12-17).

**P10L25:** Difficult to read when so many values are listed with their standard deviation.

To avoid confusion, we have deleted this sentence in the revised manuscript.

**Section 3.3:** Could you add a discussion on the SOC mass ranges using the defined uncertainties for the ratios? Is there perhaps more up-to-date values to be used?

Thanks. We have added some additional discussion. Pleased see "The average concentrations of estimated SOC before, during, and after the Parade (marked as Before-P, During-Parade, and After-P, respectively) are shown in Figure 8. The estimated SOC concentrations during the Parade ($320\pm111$ ngC m$^{-3}$, $370\pm163$ ngC m$^{-3}$ and $594\pm264$ ngC m$^{-3}$ at 8 m, 120 m and 260 m, respectively) decreased by ~12% ($364\pm199$ ngC m$^{-3}$) and 10% ($356\pm177$ ngC m$^{-3}$) at 8 m, 35% ($571\pm419$ ngC m$^{-3}$) and 16% ($441\pm279$ ngC m$^{-3}$) at 120 m; decreased 31% ($864\pm585$ ngC m$^{-3}$) and increased 4% ($570\pm229$ ngC m$^{-3}$) at 260 m when compared to the Before-P and After-P, respectively. The SOC at the upper layers decreased more than at the ground surface layer, suggesting the efficient mitigation of SOC on a regional scale. The previous studies during the

same period (Wu et al., 2019; Zhao et al., 2017) showed a high frequency of southerly winds before the Parade and north winds during the Parade at the high layers. It suggests that the north winds were also an important reason for the reduction of SOC during the Parade.

We found that the fractions of ASOC decreased and Iso_SOC increased for the emission controls. The ASOC fractions at 8 m were 59±8% (Before-P), 47±5% (During-Parade), and 57±8% (After-P), and Iso_SOC were 18±5%, 18±2%, and 12±2%, respectively. The ASOC fractions at 120 m were 64±5% (Before-P), 61±10% (During-Parade) and 65±8% (After-P), and Iso_SOC were 17±5%, 23±6%, and 16±6%, respectively. The ASOC fractions at 260 m were 63±10% (Before-P), 53±9% (During-Parade) and 64±9% (After-P), and Iso_SOC were 24±8%, 34±9% and 21±9%, respectively." in the revised manuscript (on page 11 lines 12-25).

**P11L12-L14:** To improve readability, could you compare Parade with average before and after?

Yes, we have modified these sentences. Please see "The average concentrations of estimated SOC before, during, and after the Parade (marked as Before-P, During-Parade, and After-P, respectively) are shown in Figure 8. The estimated SOC concentrations during the Parade ($320\pm111$ ngC m$^{-3}$, $370\pm163$ ngC m$^{-3}$ and $594\pm264$ ngC m$^{-3}$ at 8 m, 120 m and 260 m, respectively) decreased by ~12% ($364\pm199$ ngC m$^{-3}$) and 10% ($356\pm177$ ngC m$^{-3}$) at 8 m, 35% ($571\pm419$ ngC m$^{-3}$) and 16% ($441\pm279$ ngC m$^{-3}$) at 120 m; decreased 31% ($864\pm585$ ngC m$^{-3}$) and increased 4% ($570\pm229$ ngC m$^{-3}$) at 260 m when compared to the Before-P and After-P, respectively." in the revised manuscript (on page 11 lines 12-16). We hope it can be more readable.

***Technical comments:***

Please check section numbering, 3.1.2 is repeated, and the reason to change from 3.1.3 to 3.2 is unclear to me.

Thanks. We are sorry to make this mistake and we have modified it.

Section 3.1 is the main description of BSOA tracers and 3.2 is the description of ASOA tracers. To connect these two sections, we have added "In addition, it is important to investigate the vertical profiles of ASOA and its interactions with BSOA. ASOA is a larger contributor to the loading of SOA and the formation of air pollution in urban areas." in the revised manuscript (on page 9 line 32 to page 10 lines 1-2).

**Fig. 1:** Unclear what the authors meant by "Obvious meteorological conditions were found during the sampling period."

We have deleted this sentence.

**P2L1:** I think the authors mean "can impact radiative forcing".

Thanks. We have modified it in the revised manuscript.

**P2L6:** remove "a" between "photooxidation" and "of".

Corrected.

**P2L15:** This sentence could be review – changing CCN size also affects the radiative forcing. I suggest "…influencing the climate negatively impacting human health" given that those aspects were already described earlier.

We have modified this sentence to "changing cloud condensation nuclei (CCN) size, influencing the climate, and damaging human health" in the revised manuscript (on page 2 lines 15-16).

**P2L22:** "…events in China highlights the urgent..." & "…processes of SOA formation in the atmosphere"

Thanks. We have changed this sentence to "The high contribution of secondary aerosols to the PM pollution during haze events in China highlights the urgent need to understand the compositions and processes of SOA formation in the atmosphere" in our revised manuscript (on page 2 lines 21-22).

**P2L26:** "…urban boundary layer are lacking, …"

Corrected.

P2L30-34: Unclear the objective of this sentence, please rewrite to make it clearer.

Thanks for the reviewer's comment. We have changed this sentence to "Several field studies at the rainforest Amazon Tall Tower Observatory (ATTO) also measured the vertical gradients of VOCs. (Andreae et al., 2015; Yáñez-Serrano et al., 2018). However, vertical SOA profiles were still lacking. A previous study reported that the loading of SOA is high above the surface layer during the summer over the southeastern United States, which was potentially related to the heterogeneous chemical and gas-to-particle reactions of BVOCs oxidation products (Goldstein et al., 2009)." in the revised manuscript (on page 2 lines 29-33).

**P3L3:** "severe" instead of "serve".

Corrected.

**P3L3-6:** Please rewrite this sentence. It feels like it's repeating several times the same phrase "understand SOA formation mechanisms to improve air quality".

Thanks for the reviewer's suggestion. We have rewritten this sentence to "It is meaningful in learning the SOA properties and probing its behaviors in the atmosphere. This information also has regulatory implications for decision makers." in the revised manuscript (on page 3 lines 1-2).

**P3L8:** "emission control" & "improve" instead of "guarantee the".

Corrected.

**P3L9:** remove "the chemical behaviors and regional transport of"

Corrected.

**P3L16-17:** this sentence is unclear

Thanks. We have modified this sentence to "The influences of emission controls during the Parade period on the characteristics of SOC were also investigated." in the revised manuscript (on page 3 lines 12-13).

**P3L17:** "To the best of our knowledge".

Corrected.

**P3L19:** "megacity in China" or "Chinese megacity".

Thanks. We have modified this to "Chinese megacity" in the revised manuscript (on page 3 line 14).

**P3L24:** I missed here a more detailed description of the site location itself, such as lat/long for example.

We have added a detailed description of the sampling site. Please see Section 2.1 in the revised manuscript.

**P4L2:** Do not skip line here.

Corrected.

**P4L3:** Change Tem for T, also in the figure.

We have changed it in the text and figure.

**P4L24:** "blank"

Corrected.

**P5L8:** Do you mean something like: "Whereas the prevailing winds at 8m were either easterly or westerly, at 260 m the wind direction was dominated by northerlies."?

Yes. We have modified this sentence to "The prevailing winds at 8 m were either easterly or westerly, while at 120 m and 260 m the wind directions were dominated by northerlies" in the revised manuscript (on page 5 lines 9-10).

**P5L22:** Table S2

Thanks. We have modified it to Tables S2. Table S2 showed the significant differences of these averages, we wanted to use this table to imply if these average values have meaning on statistics.

**P6L31:** "…, while tracers of monoterpenes and sesquiterpenes SOA did not show a marked increase with height."
Corrected.

**P7L1-2:** repetition of information.

Thanks. We have modified this sentence to "…., while the fractions from monoterpene SOA tracers and sesquiterpene SOA tracer decreased from 33% to 21% and 4% to 2%, respectively (Figure 2d and Figure S7)" in the revised manuscript (on page 7 lines 19-21).

**P9L15:** Which other pollution events? I thought they were only three.

Yes. There were only three pollution events in our study. We have modified this sentence to "Total concentrations of BSOA tracers increased with height during the August 17th and 19th episodes (E1) and the August 29th episode (E2), and complex vertical distributions were recorded in other pollution days." in the revised manuscript (on page 9 lines 17-18).

**P11L12:** "Before".
Corrected.

**P11L16:** remove "obviously"
Corrected.

**P11L27:** please rephrase.

We have rephrased this sentence to "The vertical properties of SOA tracers in aerosols were investigated over the late summer in Beijing" in the revised manuscript (on page 12 line 2).

**References**

Andreae MO, Acevedo OC, Araùjo A, Artaxo P, Barbosa CGG, Barbosa HMJ, et al. The Amazon Tall Tower Observatory (ATTO): overview of pilot measurements on ecosystem ecology, meteorology, trace gases, and aerosols. Atmos. Chem. Phys. 2015; 15: 10723–10776.

Borbon A, Fontaine H, Veillerot M, Locoge N, Galloo JC, Guillermo R. An investigation into the traffic-related fraction of isoprene at an urban location. Atmos. Environ. 2001; 35: 3749–3760.

Claeys M, Graham B, Vas G, Wang W, Vermeylen R, Pashynska V, et al. Formation of secondary organic aerosols through photooxidation of isoprene. Sicence 2004; 303: 1173–1176.

Ding X, Wang XM, Zheng M. The influence of temperature and aerosol acidity on biogenic secondary organic aerosol tracers: Observations at a rural site in the central Pearl River Delta region, South China. Atmos. Environ. 2011; 45: 1303–1311.

Du W, Dada L, Zhao J, Chen X, Daellenbach KR, Xie C, et al. A 3D study on the amplification of regional haze and particle growth by local emissions. npj Climate and Atmospheric Science 2021; 4.

Du W, Zhao J, Wang YY, Zhang YJ, Wang QQ, Xu WQ, et al. Simultaneous measurements of particle number size distributions at ground level and 260 m on a meteorological tower in urban Beijing, China. Atmos. Chem. Phys. 2017; 17: 6797–6811.

Faiola CL, Vanderschelden GS, Wen M, Elloy FC, Cobos DR, Watts RJ, et al. SOA formation potential of emissions from soil and leaf litter. Environ. Sci. Technol. 2014; 48: 938–946.

Fu PQ, Kawamura K, Chen C, Barrie LA. Isoprene, Monoterpene, and Sesquiterpene Oxidation Products in the High Arctic Aerosols during Late Winter to Early Summer. Environ. Sci. Technol. 2009; 43: 4022–4028.

Goldstein AH, Koven CD, Heald CL, Fung IY. Biogenic carbon and anthropogenic pollutants combine to form a cooling haze over the southeastern United States. P. Natl. Acad. Sci. USA. 2009; 106: 8835–8840.

Li J, Du H, Wang Z, Sun Y, Yang W, Li J, et al. Rapid formation of a severe regional winter haze episode over a mega-city cluster on the North China Plain. Environ. Pollut. 2017; 223: 605-615.

Miao Y, Hu X-M, Liu S, Qian T, Xue M, Zheng Y, et al. Seasonal variation of local atmospheric circulations and boundary layer structure in the Beijing-Tianjin-Hebei region and implications for air quality. J. Adv. Model. Earth. SY. 2015; 7: 1602-1626.

Shilling JE, Zaveri RA, Fast JD, Kleinman L, Alexander ML, Canagaratna MR, et al. Enhanced SOA formation from mixed anthropogenic and biogenic emissions during the CARES campaign. Atmos. Chem. Phys. 2012; 12: 26297–26349.

Shrivastava M, Andreae MO, Artaxo P, Barbosa HMJ, Berg LK, Brito J, et al. Urban pollution greatly enhances formation of natural aerosols over the Amazon rainforest. Nat. Commun. 2019; 10: 1046.

Sun Y, Du W, Wang Q, Zhang Q, Chen C, Chen Y, et al. Real-time characterization of aerosol particle composition above the urban canopy in Beijing: insights into the interactions between the atmospheric boundary layer and aerosol chemistry. Environ. Sci. Tech. 2015; 49: 11340–7.

Szmigielski R, Surratt JD, Gómez-González Y, Van der Veken P, Kourtchev I, Vermeylen R, et al. 3-methyl-1,2,3-butanetricarboxylic acid: An atmospheric tracer for terpene secondary organic aerosol. Geophys. Res. Lett. 2007; 34: L24811.

Wang HC, Lu KD, Chen XR, Zhu QD, Wu ZJ, Wu YS, et al. Fast particulate nitrate formation via N2O5 uptake aloft in winter in Beijing. Atmos. Chem. Phys. 2018a; 18: 10483–10495.

Wang Q, Sun Y, Xu W, Du W, Zhou L, Tang G, et al. Vertically resolved characteristics of air pollution during two severe winter haze episodes in urban Beijing, China. Atmos. Chem. Phys. 2018b; 18: 2495–2509.

Wang Q, Zhang Q, Ma Z, Ge B, Xie C, Zhou W, et al. Temporal characteristics and vertical distribution of atmospheric ammonia and ammonium in winter in Beijing. Sci. Total. Environ. 2019; 681: 226-234.

Wang W, H. WM, Li L, Zhang T, Liu XD, Feng JL, et al. Polar organic tracers in PM2.5 aerosols from forests in eastern China. Atmos. Chem. Phys. 2008; 8: 7507–7518.

Wei LF, Yue SY, Zhao WY, Yang WY, Zhang YJ, Ren LJ, et al. Stable sulfur isotope ratios and chemical compositions of fine aerosols (PM 2.5 ) in Beijing, China. Sci. Total. Environ. 2018; 633: 1156–1164.

Wu L, Ren H, Wang P, Chen J, Fang Y, Hu W, et al. Aerosol ammonium in the urban boundary layer in Beijing: insights from nitrogen isotope ratios and simulations in summer 2015. Environ. Sci. Tech. Let. 2019; 6: 389–395.

Yáñez-Serrano AM, Nölscher AC, Bourtsoukidis E, Gomes Alves E, Ganzeveld L, Bonn B, et al. Monoterpene chemical speciation in a tropical rainforest:variation with season, height, and time of dayat the Amazon Tall Tower Observatory (ATTO). Atmos. Chem. Phys. 2018; 18: 3403–3418.

Zelenyuk A, Imre DG, Wilson J, Bell DM, Suski KJ, Shrivastava M, et al. The effect of gas-phase polycyclic aromatic hydrocarbons on the formation and properties of biogenic secondary organic aerosol particles. Faraday Discuss 2017; 200: 143–164.

Zhao J, Du W, Zhang YJ, Wang QQ, Chen C, Xu WQ, et al. Insights into aerosol chemistry during the 2015 China Victory Day parade: results from simultaneous measurements at ground level and 260 m in Beijing. Atmos. Chem. Phys. 2017; 17: 3215–3232.

Zheng GJ, Duan FK, Su H, Ma YL, Cheng Y, Zheng B, et al. Exploring the severe winter haze in Beijing: the impact of synoptic weather, regional transport and heterogeneous reactions. Atmos. Chem. Phys. 2015; 15: 2969–2983.

---

## Author Comment (AC2)

**Responses to Referee 2**

We are thankful to the reviewer for his/her thoughtful comments and suggestions.
We have revised the manuscript accordingly. Listed below are our point-by-point responses in blue to the comments. The modified parts in the revised manuscript are highlighted in yellow.

**RC2**: 'Comment on acp-2021-136', Anonymous Referee #2, 31 Mar 2021 reply
The authors present measurements of aerosol mass and composition (i.e., tracer concentrations) at a tall tower in Beijing. Measurements of vertical distributions, as presented here, are valuable and generally fairly scarce, so presenting these measurements is itself of value to the community. I believe the work is useful and worth publishing in this journal, but suffers from some scientific overreach that needs to be addressed first. Specifically, as described below, the authors need to temper the strength of some of their statements to more accurately reflect the strength of their evidence, and the authors need to re-evaluate some of their interpretation of tracer ratios by either providing support from the literature or correcting their claims.
We thank the reviewer's comments and valuable suggestions. All comments and suggestions have been considered carefully and addressed below.

**General comments:**
1. It is a little confusing that the methods discuss Parade-based periods, but most of the paper actually is more about the pollution episodes.
Thanks for the reviewer's comments. We have discussed the properties of SOA tracers during the pollution events in Section 3.1 and the impacts of emission controls on SOC in Section 3.4 in the revised manuscript. We think that the reduction measurements during the Parade provide a unique chance to study SOA under the government interventions as mentioned in the last paragraph in Section 1. These results are conducive to the making of reduction policies.

2. There is a fair amount of English-language issues, mostly odd phrasing and the like, that should be cleaned up.
Thanks. We have revised the language of our manuscript that are highlighted in yellow. We hope the revised manuscript can meet the requirement of the journal.

3. Interpretation of tracer data is somewhat confusing and does not seem accurate to me, particularly in the case of the isoprene tracers. In particular, the authors' interpretation of the 2-MT/2-MET and the 2-MTs/C5-ATs ratio are not, to the best of my knowledge, grounded in recent literature on the sources of these tracers, and the citations provided by to the authors do not support their interpretations as far as I can tell.

Thanks for the reviewer's comments. We have rewritten Section 3.1.3 in the revised manuscript. We have deleted the discussion of 2-MT/2-MET and added some recent references about the ratio of 2-MTs/$C_5$-alkene triols. We hope these modifications can improve the quality of our manuscript.

4. The authors seem to draw fairly broad conclusions from somewhat limited evidence. While the vertical distributions are certainly interesting, some of the assumptions regarding regional vs. local transport, changes in partitioning, impacts of primarily particles, etc., are not very strongly supported. In some cases, these claims seem to based off of previous work, but that is not always clear. In other cases, these claims are based on tracer ratios, but as described above, it's not clear these claims are always based on a current understanding of the current tracer literature.

We have noticed the confusion of some claims in our study, and we have rewritten these claims with a clearer expression. In addition, as mentioned above, we have rewritten our description of tracer ratios and added some current literature in the revised manuscript. Please see these changes in Section 3.1 in our revised manuscript. We hope these changes can meet the high standard of the journal.

**Specific comments:**

P1L29: This is a very broad statement, which is fine as an opening sentence, but why are these citations specifically selected? Some of the early ones make sense, but, for instance, what is the information being conveyed by the Huang et al. reference?

We are sorry that we have cited this reference in an improper place. We have deleted it and we have checked other references in our manuscript. We hope that our manuscript has no this problem again.

P2L6: extra "a"

Thanks. We have deleted it in the revised manuscript

P2L9-10: This distinction between ASOA and BSOA is a bit out of date, in the years after these citations the idea of ABSOA being dominant became somewhat more accepted. The next sentence clarifies this a bit, but the statement that 90% of SOA is BSOA is more a historical perspective than actually informative so should be re-framed as such or removed.

Thanks for the comments and suggestions of the reviewer. We have re-framed this sentence to "Anthropogenic SOA (ASOA) and biogenic SOA (BSOA) are important contributors to OA and air pollution in the atmosphere (Huang et al., 2014; Volkamer et al., 2006)" in the revised manuscript (on page 2 lines 10-11).

P3L6: should say decision "makers"

Thanks. We have modified it.

P3L12: What is the basis for the claim that 120 m and 260 m are regional? There are two citations - do they measure boundary layer height? Or model it? Or measure tracer compounds in some way?

These two citations (Sun et al., 2015; Zhao et al., 2020) have measured tracer compounds and showed profiles of meteorological conditions based on the 325-m meteorological tower in the boundary layer. These results highlight that the measurements at 260 m are more representative of regional sources whereas the ground measurements are more subject to local sources. Hence, the claim here was based on the citations.

P3L27: I'm not sure "the typical urban site" really means anything. The following description of the site is more useful.

We have modified this to "an urban site" and added more descriptions about the sampling site. Please see "The sampling site is at the Institute of Atmospheric Physics (IAP), Chinese Academy of Sciences (39°58.53′N, 116°22.69′E), which is in an urban site (between 3- and 4-ring) of Beijing and surrounded by street road (~50 m), highway (~300 m), a public park (~500 m to the southwest), restaurants (~100 m), residential housing and a gas station (~200 m). The predominant vegetation types surrounding the sampling site are deciduous broadleaf vegetation (acacia and juglandaceae), shrub, and lawn. The vegetation cover of the public park is more than 50%. The predominant vegetation is also deciduous broadleaf." in Section 2.1 in the revised manuscript (on page 3 liens 23-28).

P4L6: "OC and EC in an aliquot filter" is phrased oddly and should be re-stated

We have rewritten this phrase to "OC and EC in aerosols" in the revised manuscript (on page 4 line 4).

P4L17: Are the author's sure it is a Hewlett-Packard? The last HP GC I was aware of, at least in the United States, was the 5890, and I thought subsequent GCs and MSs were all sold under the "Agilent" branding (i.e., Agilent 7890 GC and Agilent 5975 MS). But perhaps it is different in other countries?

Thanks for the comments of the reviewer. Yes, our instrument was bought from "Agilent" branding, but the mainboard is Hewlett-Packard. We have modified it to "a Hewlett-Packard model Agilent 7890A GC coupled to Hewlett-Packard model Agilent 5975C mass selective detector (MSD)." in the revised manuscript (on page 4 lines 15-16).

P4L24: Were they not corrected for recoveries because recoveries was near 100%? That should be stated if so.

Many SOA standards are not commercially available. As mentioned in the reference (Fu et al., 2009), only several SOA standards are available, other SOA tracers were obtained by comparison with those of literature data or by using the surrogates. Hence, we did not add standards in our laboratory into the samples, we only added the internal standard $C_{13}$ n-alkane. It is why recoveries were not used for correction in our manuscript.

P5L20-23: These sentences seem to contradict, claiming both increase with height and no significant differences with height.

To avoid ambiguity, we have modified this sentence to "The concentrations of WSOC and OC were 2.73 ± 1.31 μgC m$^{-3}$ and 5.03 ± 2.28 μgC m$^{-3}$ at 260 m, 2.69 ± 1.55 μg m$^{-3}$ and 5.32 ± 2.88 μg m$^{-3}$ at 120 m, and 2.03 ± 0.99 μg m$^{-3}$ and 4.37 ± 1.69 μg m$^{-3}$ at 8 m, respectively" in the revised manuscript (on page 5 lines 23-24).

P527-30: Sometimes it is not clear to me when the authors are making a new claim, vs. stating a previously published result. This statement is one of those examples - is the claim that the lower WSOC:OC ratio at ground level is due to biological aerosols a claim made (and presumably supported) by Wang et al., or is that a new claim here?

We are sorry for this confusion. It is a new claim, but we used a confusing way of writing. We have changed this sentence to "In addition, primary sources from local dust and soil resuspension, such as primary biological aerosols which contain a high abundance of water-insoluble organic compounds (Wang et al., 2019), potentially caused the lower fractions of WSOC to OC at the ground surface than at the upper layers." in the revised manuscript (on page 5 line 31 to page 6 lines 1-3). We also revised the same problem in our manuscript, we hope these changes can give a clearer expression for the claims.

P6L4 and P6L29: This sentence is a bit misleading - some monoterpene tracers decrease, but others (MBTCA, HDCCA) increase. Also, there is only one sesquiterpene tracer, so it is a bit tough to make general claims like this.

Thanks for the reviewer's comments. We have modified our expression to "Most of these molecular tracers showed higher abundance at high layers (≥ 120 m) than at 8 m, except for pinic acid, pinonic acid, 3-acetuldipic acid and β-caryophyllinic acid. Table S2 shows significant differences in the average concentrations of these SOA tracers with height, except for monoterpene SOA tracers." in the revised manuscript (on page 6 lines 6-9). We hope this expression can be more appropriate.

P7L4: The claim that isoprene is regional and MTs/SQTs are more local is not necessarily true. As the authors note, the vertical distribution could be due to regional transport, but conversely could be due to vertical differences in chemistry and/or partitioning.

Thanks for the valuable suggestions of the reviewer. We have changed this claim to "It suggests that regional transport potentially contributes more to isoprene SOA, while SOA from monoterpenes and sesquiterpene is likely more influenced by local sources. In addition, some other factors (such as transformation and condensation processes) can also lead to these patterns." in the revised manuscript (on page 7 lines 22-24). We hope this expression can be more appropriate.

P7L26: Is there a citation for the claim that sesquiterpenes are mainly emitted by crops and herbs? I'm not sure that is true, they are released from many plants, particularly for reasons related to chemical signaling and plant protection (e.g., increase SQTs with herbivory: Faiola et al, 10.1021/acsearthspacechem.9b00118)

Thanks for the comments of the reviewer. A previous study has mentioned that "The SQT emissions distribution is strongly influenced by the grass and crop PFT." (Sakulyanontvittaya et al., 2008). However, it was not suitable to claim that sesquiterpenes are mainly emitted by crops and herbs in our manuscript. We are very thankful for the remainder of the reviewer. Many factors can influence the emission of sesquiterpenes (Duhl et al., 2008; Faiola et al., 2019). Hence, we have modified our claim to "Sesquiterpenes are mainly emitted from plants and trees, which are controlled by many factors, such as temperature and stage of plant growth (Duhl et al., 2008; Faiola et al., 2019)." in the revised manuscript (on page 6 lines 26-28). We have deleted the explain here in the original manuscript.

P8L4: The phase" methacryloyl peroxynitrate (MPAN, e.g. methacrolein, methyl vinyl ketone and methyl butanediols)" is odd, as those latter species are not a subset of MPAN but rather separate compounds

Thanks for the reminder of the reviewer. We have added new references and modified this sentence to "The isoprene oxidation mechanisms are dependent on atmospheric conditions (Bates and Jacob, 2019; Wennberg et al., 2018)." in our revised manuscript (on page 8 lines 1-2).

P8L7: I find the use of 2-MT to mean 2-methylthreitol while 2-MTs means the sum of both isomers to be confusing. I would recommend calling the sum 2-MTs (which is fairly standard) and maybe calling the isomers 2-MT_eryth and 2-MT_threi (where "_X" denotes a subscript).

Thanks. According to the reviewer's suggestion, we have changed the abbreviation of these two isomers in our revised manuscript.

P8L16: I am not aware of work showing that the 2-MT/2-MET ratio is indicative of anything in particular. The two citations in this sentence do not seem to include such claims either. As someone who has thought a fair amount about isoprene and monoterpene tracers, it's not clear to me what this ratio is telling me, or why the authors include it.

Thanks for the reviewer's comments. We have detected the discussion of 2-MT/2-MET ratio and modified figure 5 in our revised manuscript.

[Figure]

Figure 5. Temporal variations in the mass concentration ratios among different biogenic SOA tracers in $PM_{2.5}$: (a) 2-MTs / 2-MGA; (b) 2-MTs / $C_5$-alkene triols and (c) MBTCA / (PA+PNA).

P8L20-23: The interpretation of C5-alkene triols as precursors in the oxidation of 2-MTs is confusing to me, to the point of making me feel the authors are interpreting their tracer data through an outdated lens. Since the Wang et al., 2005 paper, lots of work has been done on IEPOX oxidation pathways, and I'm not aware that any of it has made the claim the authors are making here. Even in the Wang et al., 2005 paper, Scheme 1 shows both C5-ATs and 2-MTs to be products of IEPOX (one through addition and one through rearrangement). Since then, there has been a fair amount of work to understand what C5-alkene triols are actually "telling us", in particular from the Surratt group and Goldstein group, and I think both groups would agree it's still not quite clear. See for example: Cui et al. doi.org/10.1039/C8EM00308D and Yee et al. 10.1021/acs.est.0c00805.

Thanks for the reviewer's comments. We have searched the references mentioned by the reviewer. The formations of $C_5$-alkene and 2-MTs are not quite clear, and many factors (such as aerosol acidity and humidity) can influence the ratios of 2-MTs/$C_5$-alkene (Cui et al., 2018; Surratt et al., 2010; Yee et al., 2020). Hence, we have rewritten the paragraph about 2-MTs and $C_5$-alkenes. Please see "The average ratios of 2-MTs to $C_5$-alkene triols were 0.97±1.17, 1.33±1.24, and 3.97±3.08 at 8 m, 120 m, and 260 m, respectively (Figure 5b). $C_5$-alkene triols have been suggested to convert into 2-MTs (Wang et al., 2005). Some studies also suggested that the loading of 2-MTs increased with the enhancement of aerosol acidity (Surratt et al.,

2007), and the relative humidity can affect the ratio of 2-MTs to $C_5$-alkene triols (Surratt et al., 2010). Recent studies suggested the ratio of 2-MTs / $C_5$-alkene triols decreased with aerosol acidity (Yee et al., 2020), and $C_5$-alkene triols were likely formed from thermal degradation of 2-methyltetrol sulfates for GC/MS artifacts (Cui et al., 2018). Hence, it is still not clear the meaning of the ratio 2-MTs to $C_5$-alkene triols. However, the large differences of 2-MTs / $C_5$-alkene triols values at three heights highlight the significance of studying vertical profiles of SOA, and more field investigations are needed." in the revised manuscript (on page 8 lines 15-22).

P8L23: Typo: "vitations"
Corrected.

P10L28: It would be helpful in the figures and pie charts about source apportionment if they also included what fraction of OC and/or WSOC was not captured by the source apportionment. I think this sentence here is telling me that only 8-13%% of SOC is accounted for in their source apportionment, but it's not totally clear to me.
Thanks for the reviewer's comments. This sentence tells the fractions of estimated SOC in OC. We have added the uncaptured fraction of OC in figure 7 in our revised manuscript according to the suggestion of the reviewer.

[Figure]

Figure 7. Temporal variations in the estimated SOC and other OC at three heights: (a) the concentrations of estimated SOC (right axis) and other OC (left axis), (b) the fraction of estimated SOC and other OC in OC. Relative mass fractions of OC and estimated SOC is shown in (c) and (d). Other OC is not captured by the source apportionment. Iso_SOC, Mon_SOC, and Sesq_SOC represent BSOC estimated from isoprene, monoterpenes, and sesquiterpene, respectively. Toluene SOC and naphthalene SOC represent ASOC that were estimated from DHOPA and phthalic acid, respectively.

P11, Sect. 3.4: Are these reductions in total WSOC, or just the fraction of SOC that is captured in the source apportionment?

Thanks. These reductions are the fraction of SOC that is captured in the source apportionment.

**Figures:**

HDCCA is not defined anywhere in the main text

We defined HDCCA in table S1 in the support information. HDCCA is the abbreviation of 3-(2-hydroxyethyl)-2,2-dimethyl-cyclobutane carboxylic acid. We have added this sentence in the caption of Figure 4 in the revised manuscript.

Figure 5. The caption is in the wrong order, and the description of panel (d) is tough to understand.

Thanks. We have changed the order and the description of panel (d). The revised caption is "Figure 5. Temporal variations in the mass concentration ratios among different biogenic SOA tracers in $PM_{2.5}$: (a) 2-MTs / 2-MGA; (b) 2-MTs / $C_5$-alkene triols and (c) MBTCA / (PA+PNA)." In the revised manuscript.

**References**

Bates KH, Jacob DJ. A new model mechanism for atmospheric oxidation of isoprene: global effects on oxidants, nitrogen oxides, organic products, and secondary organic aerosol. Atmos. Chem. Phys. 2019; 19: 9613-9640.

Cui T, Zeng Z, Santos EOd, Zhang Z, Chen Y, Zhang Y, et al. Development of a hydrophilic interaction liquid chromatography (HILIC) method for the chemical characterization of water-soluble isoprene epoxydiol (IEPOX)-derived secondary organic aerosol. Environ. Sci.: Processes Impacts 2018; 20: 1524-1536.

Duhl TR, Helming D, Guenther A. Sesquiterpene emissions from vegetation: a review. Biogeosciences. 2008; 5: 761–777.

Faiola CL, Pullinen I, Buchholz A, Khalaj F, Ylisirnio A, Kari E, et al. Secondary Organic Aerosol Formation from Healthy and Aphid-Stressed Scots Pine Emissions. ACS Earth Space Chem 2019; 3: 1756-1772.

Fu PQ, Kawamura K, Chen C, Barrie LA. Isoprene, Monoterpene, and Sesquiterpene Oxidation Products in the High Arctic Aerosols during Late Winter to Early Summer. Environ. Sci. Technol. 2009; 43: 4022–4028.

Huang RJ, Zhang YL, Bozzetti C, Ho KF, Cao JJ, Han Y, et al. High secondary aerosol contribution to particulate pollution during haze events in China. Nature 2014; 514: 218–222.

Sakulyanontvittaya T, Duhl T, Wiedinmyer C, Helmig D, Matsunaga S, Potosnak M, et al. Monoterpene and sesquiterpene emission estimates for the United States. Environ. Sci. Technol. 2008; 42: 1623-1629.

Sun Y, Du W, Wang Q, Zhang Q, Chen C, Chen Y, et al. Real-time characterization of aerosol particle composition above the urban canopy in Beijing: insights into the interactions between the atmospheric boundary layer and aerosol chemistry. Environ. Sci. Tech. 2015; 49: 11340–7.

Surratt JD, Chan AW, Eddingsaas NC, Chan M, Loza CL, Kwan AJ, et al. Reactive intermediates revealed in secondary organic aerosol formation from isoprene. P. Natl. Acad. Sci. USA. 2010; 107: 6640–6645.

Surratt JD, Lewandowski M, Jaoui M, Kleindienst TE, Edney EO, Seinfeld JH. Effect of acidity on secondary organic aerosol formation from isoprene. Environ. Sci. Technol. 2007; 41: 5363–5369.

Volkamer R, Jimenez JL, San Martini F, Dzepina K, Zhang Q, Salcedo D, et al. Secondary organic aerosol formation from anthropogenic air pollution: Rapid and higher than expected. Geophys. Res. Lett. 2006; 33 (17): L17811.

Wang S, Song T, Shiraiwa M, Song J, Ren H, Ren L, et al. Occurrence of aerosol proteinaceous matter in urban Beijing: an investigation on composition, sources, and atmospheric processes during the "APEC Blue" period. Environ. Sci. Technol. 2019; 53: 7380–7390.

Wang W, Kourtchev I, Graham B, Cafmeyer J, Maenhaut W, Claeys M. Characterization of oxygenated derivatives of isoprene related to 2-methyltetrols in Amazonian aerosols using trimethylsilylation and gas chromatography/ion trap mass spectrometry. Rapid Comun. Mass Sp. 2005; 19: 1343–1351.

Wennberg PO, Bates KH, Crounse JD, Dodson LG, McVay RC, Mertens LA, et al. Gas-Phase Reactions of Isoprene and Its Major Oxidation Products. Chem. Rev. 2018; 118: 3337-3390.

Yee LD, Isaacman-VanWertz G, Wernis RA, Kreisberg NM, Glasius M, Riva M, et al. Natural and Anthropogenically Influenced Isoprene Oxidation in Southeastern United States and Central

Amazon. Environ. Sci. Technol. 2020; 54: 5980-5991.

Zhao WY, Ren H, Kawamura K, Du HY, Chen XS, Yue SY, et al. Vertical distribution of particle-phase dicarboxylic acids, oxoacids and alpha-dicarbonyls in the urban boundary layer based on the 325m tower in Beijing. Atmospheric Chemistry and Physics 2020; 20: 10331-10350.

---

## Editor Decision (ED1)

Editor report on the revised version of acp-2021-136 "Measurement report: Vertical distribution of biogenic and anthropogenic secondary organic aerosols in the urban boundary layer over Beijing during late summer"

Line numbers refer to the annotated 'track-change' manuscript.

p. 1, l. 31: replace 'undertaking' by 'undertaken'

p. 2, l. 2: replace 'radiation force' by 'radiative forcing'

p. 2, l. 10ff: These references seem somewhat outdated and should be amended by more recent ones. Please take a look at more recent ones and select those that are relevant to your discussion and should be included, e.g. – but not limited to - (Hodzic et al., 2016, 2020; Huang et al., 2020; Liu et al., 2021; Pai et al., 2020)

p. 2, l. 32: 'heterogeneous chemical and gas-to-particle reactions' should be replaced by 'heterogeneous chemical reactions and gas-to-particle conversion'

p. 3, l. 17: Add 'good' or 'improved' before 'air quality'

p. 5, l. 11: '…are about half to several hundred meters away from the sampling site.' – It is not clear what 'half' relates to in this context.

p. 5, l. 27: 51.1%% - redundant '%' sign

p. 6, l. 5: Define DHOPA

p. 6, l. 7: '3-acetuldipic acid' – please clarify this species name. It does not show up in Tables S1 or S2 and does not seem to be a common name. I assume you might mean acetyl adipic acid? (If so, please also correct Figure 4k)

p. 8, l. 15-22: I am not sure that I understand this new text, in particular the last sentence. If conclusions based on the 2-MT/triol ratio as derived from lab measurements are rather ambiguous, why do the large differences in observed ratios here lead to the conclusion that more vertical SOA measurements are needed? Obviously the 2-MT/triol ratio is not useful in concluding on SOA formation and modification processes.

p. 9, l. 15: 'formation of E2' should be 'air masses during E2' (or similar)

p. 10, l. 2: Please add a reference to the statement "ASOA is a larger contributor to the loading of SOA and the formation of air pollution in urban areas."

p. 10, l. 14: Do you mean 'biogenic aerosols' by 'natural aerosols'?

p. 10, l. 14-17: It is not clear how this text relates to your findings. Did you also observe higher BSOA formation when anthropogenic factors were enhanced? Are you referring to effects such as enhanced BSOA yields at high NOx levels or similar? Make the connection of the literature studies to your findings here clearer.

p. 11, l. 21: 'We found that the fractions of ASOC decreased and Iso_SOC increased for the emission controls' – This sentence is not clear. Should it read '…as a response to the emission controls'? Please clarify.

Data availability: Please note our guidelines for Measurement Reports that state that 'The data presented in measurement reports must be openly accessible in accordance with the EGU data policy.' https://www.atmospheric-chemistry-and-physics.net/about/manuscript_types.html

Please make the data available in a public data repository https://www.atmospheric-chemistry-and-physics.net/policies/data_policy.html

---

## Author Response (AR2)

**Responses to Editor**

We are thankful to the editor for the thoughtful comments and suggestions.

We have revised the manuscript accordingly. Listed below are our point-by-point responses in blue to the comments. The modified parts in the revised manuscript are highlighted in yellow.

Editor report on the revised version of acp-2021-136 "Measurement report: Vertical distribution of "biogenic and anthropogenic secondary organic aerosols in the urban boundary layer over Beijing during late summer"

Line numbers refer to the annotated 'track-change' manuscript.

We thank the editor's thoughtful comments. All suggestions have been addressed below.

p. 1, l. 31: replace 'undertaking' by 'undertaken'

Modified. (on page 1 line 31)

p. 2, l. 2: replace 'radiation force' by 'radiative forcing'

Modified. (on page 2 line 2)

p. 2, l. 10ff: These references seem somewhat outdated and should be amended by more recent ones. Please take a look at more recent ones and select those that are relevant to your discussion and should be included, e.g. – but not limited to - (Hodzic et al., 2016, 2020; Huang et al., 2020; Liu et al., 2021; Paiet al., 2020)

We have amended the references. Please see "(An et al., 2019; Hodzic et al., 2016; Nault et al., 2021)." in the revised manuscript. (on page 3 line 11)

p. 2, l. 32: 'heterogeneous chemical and gas-to-particle reactions' should be replaced by 'heterogeneous chemical reactions and gas-to-particle conversion'

Modified. (on page 2 lines 32-33)

p. 3, l. 17: Add 'good' or 'improved' before 'air quality'

Modified. (on page 3 line 17)

p. 5, l. 11: '…are about half to several hundred meters away from the sampling site.' – It is not clear what 'half' relates to in this context.

We have deleted "half to" and changed this sentence to "Some high buildings are several hundred meters away from the sampling site" in the revised manuscript. (on page 5 lines 11-12)

p. 5, l. 27: 51.1%% - redundant '%' sign

Modified. (on page 5 line 26)

P. 6, l. 5: Define DHOPA

Thanks. DHOPA is the abbreviation of 2,3-Dihydroxy-4-oxopentanoic acid. We have defined DHOPA in the revised manuscript. (on page 6 line 4)

p. 6, l. 7: '3-acetuldipic acid' – please clarify this species name. It does not show up in Tables S1 or S2 and does not seem to be a common name. I assume you might mean acetyl adipic acid? (If so, please also correct Figure 4k)

Sorry for the mistake. We have changed this word to "3-acetyladipic acid" in the manuscript. (on page 6 line 7). We have also changed Figure 4k. (on page 21)

p. 8, l. 15-22: I am not sure that I understand this new text, in particular the last sentence. If conclusions based on the 2-MT/triol ratio as derived from lab measurements are rather ambiguous, why do the large differences in observed ratios here lead to the conclusion that more vertical SOA measurements are needed? Obviously the 2-MT/triol ratio is not useful in concluding on SOA formation and modification processes.

Thanks for the editor's comments. The ratios of 2-MTs/$C_5$-alkene triols from lab measurements are not very useful for concluding SOA formation and modification processes. This discussion in our work is to express the obvious difference of 2-MTs to $C_5$-alkene triols at three heights and explain the reasons for this difference. However, we didn't find reasonable reasons to explain this result. We have modified this sentence "Hence, it is still not clear the meaning of the ratio 2-MTs to C5-alkene triols. However, the large differences of 2-MTs / C5-alkene triols values at three heights highlight the significance of studying vertical profiles of SOA, and more field investigations are needed." to "Hence, it is difficult to understand the different ratios of 2-MTs to $C_5$-alkene triols at three heights, and more filed investigations on the vertical profiles of SOA are needed." (on page 8 lines 19-20)
We hope this expression can be clear to the readers.

p. 9, l. 15: 'formation of E2' should be 'air masses during E2' (or similar)

This sentence aims to explain the formation of E2. We have changed this to "the E2 is largely influenced by regional transport" in the revised manuscript. (on page 9 line 12)

p. 10, l. 2: Please add a reference to the statement "ASOA is a larger contributor to the loading of SOA and the formation of air pollution in urban areas."

We have added two references "(An et al., 2019; Fan et al., 2020)" in the revised manuscript. (on page 10 line 31)

p. 10, l. 14: Do you mean 'biogenic aerosols' by 'natural aerosols?

The "natural aerosols" was used in the reference (Shrivastava et al., 2019), so we used natural aerosols in the manuscript.

p. 10, l. 14-17: It is not clear how this text relates to your findings. Did you also observe higher BSOA formation when anthropogenic factors were enhanced? Are you referring

to effects such as enhanced BSOA yields at high NOx levels or similar? Make the connection of the literature studies to your findings here clearer.

Thanks for the editor's comments. We found that the anthropogenic tracer DHOPA showed moderate correlations with some biogenic SOA tracers. The concentrations of some BSOA and ASOA tracers increased simultaneously (Figure S6). We referred to these literature studies to suggest the interactions between ASOA and BSOA.

According to the editor's comments, it is likely improper expression. To make the connection of the literature studies to the results in our study clearer, we changed these sentences to "Previous studies have reported that urban pollution can enhance the formation of natural aerosols (Shrivastava et al., 2019); the existence of aromatic compounds can lead to high loading of α-pinene-derived SOA (Shilling et al., 2012; Zelenyuk et al., 2017). These moderate correlations also suggest that the anthropogenic sources are related to biogenic sources, and their interaction mechanisms still need more investigation." in the revised manuscript. (on page 10 lines 12-15)

p. 11, l. 21: 'We found that the fractions of ASOC decreased and Iso_SOC increased for the emission controls' – This sentence is not clear. Should it read '…as a response to the emission controls'? Please clarify.

We have modified this sentence to "We found that the fractions of ASOC decreased and Iso_SOC increased as a response to the emission controls." in the revised manuscript. (on page 11 line 19)

Data availability: Please note our guidelines for Measurement Reports that state that 'The data presented in measurement reports must be openly accessible in accordance with the EGU data policy.'
https://www.atmospheric-chemistry-and-physics.net/about/manuscript_types.html
Please make the data available in a public data repository
https://www.atmospheric-chemistry-and-physics.net/policies/data_policy.html

We have added a way to get the dataset. Please see "The atmospheric particulate matter data used for analysis are available in the Supplementary Material, and the data are also available upon request from the corresponding author Pingqing Fu (fupingqing@tju.edu.cn). The dataset can also be found online (at https://10.11922/sciencedb.00069). The DOI will be valid automatically as soon as ScienceDB publishes." in the revised manuscript. (on page 12 lines 22-25).

**Responses to Referee 2**

The authors have fairly thoroughly addressed the concerns raised in my previous review. I find the revised manuscript more compelling in terms of the conclusions it draws regarding regional vs. local transport. There are a few relatively minor issues noted below, which should be addressed prior to publication.

We are thankful to the reviewer for the thoughtful comments and suggestions.

There remain some typos and odd English, As far as I can tell, none of the authors are from an English-speaking institution, so that is understandable, but it might be useful to a service proofread for English. Examples of these issues include:

The language of this manuscript has been edited by International Science Editing (http://www.internationalscienceediting.com).

P1L31: "have been undertaking"

We have corrected it to "have been undertaken". (on page 1 line 31)

P3L1: "It is meaningful in learning"

We have corrected it to "It is meaningful to learn the SOA properties and probe its behaviors in the atmosphere.". (on page 2 line 31)

P3L25: "street road"

We have corrected it to "traffic road". (on page 3 line 25)

P5L11: "half to several hundred meters away"

We have corrected it to "several hundred meters away from the sampling site". (on page 5 line 11-12)

P6L29: "Terrestrial plantations"

We have corrected it to "Terrestrial vegetation". (on page 6 line 28)

P7L11-12: "concentrations of ... concentrations"

We have deleted the second concentrations.

P7L28: "influenced by multi-factors"

We have corrected it to "many causes". (on page 7 line 28)

Minor comments:

P2L3: "IPCC", though a well-known abbreviation should probably still be defined. Similarly "the IPCC report" is fairly informal. Probably the report itself should just be cited, as opposed to a link to the IPCC website.

Thanks for the reviewer. We have defined "IPCC" and modified this sentence to "these

impacts were all shown in the Intergovernmental Panel on Climate Change (IPCC) report (IPCC, 2014)." (on page 2 lines 3-4)

P3L24-25: The phrase "(between 3- and 4-ring)" would probably be very confusing to someone lacking knowledge of Beijing's layout. Maybe changed to "(between the 3rd and 4th Ring Road)" or something like that.
We have modified it to "between the 3rd and 4th ring Road". (on page 3 lines 24-25)

P5L26: The authors use WSOC and OC concentrations to determine that aerosol are well mixed, then go on to describe differences in composition at each height. Wouldn't differences in composition imply that the aerosols are indeed not completely well mixed?
Thanks for the reviewer's comments. We seriously considered the question of the reviewer. We think that it is improper to determine the mixed properties of the aerosols with the average concentrations of WSOC and OC. Hence, we deleted this sentence "indicating that aerosols were well mixed within the boundary layer" in the revised manuscript.

P7L5: The increase in concentration is not uniform, it is driven by some of the species (particularly 2-MTs, I think). It is thus a bit odd to discuss it as a whole since the reasons for it are potentially specific to those species. For instance, if it is 2-MTs (which are almost entirely particle phase) regional transport is likely more dominant, but if it is driven by pinic acid (which partitions based on vapor pressure) then the lower temperatures may be more important.
Thanks for the reviewer's comments. We can feel the reviewer's rigorous attitude towards research. As mentioned by the reviewer, it is a bit inappropriate simply to discuss the vertical properties of the average concentrations. Hence we have changed this sentence "The increase of their concentration with height is potentially linked to the regional transport and gas-to-particle processes of semi-volatile VOCs due to lower temperatures at the upper layers (Goldstein et al., 2009). Moreover, vertical convection transport and BVOCs emission sources cannot be ignored" to "The vertical distribution properties of BSOA tracers are related to complicated factors, such as regional transport and ambient temperatures influencing different BSOA species (Goldstein et al., 2009)" in the revised manuscript. (on page 7 lines 3-4 )

P7L9: My original suggestions for 2-MT_eryth were not literal, rather I was trying to convey that "eryth" would be a subscript (but indicating this is limited by the textbox into which I enter my report). Using "_eryth" is fairly non-standard, though could be used if the editor is okay with it.
We have changed "eryth" and "threi" to subscript. Please see "2-MT$_{eryth}$" and "2-MT$_{threi}$" in the revised manuscript.

P8L13: I don't know the work of Ding et al., but are the authors certain quantification was performed the same way? A difference in 2-MT or 2-MGA quantification (for which authentic standards are lacking) could bias the ratio. It wouldn't impact the

conclusions drawn within this dataset, but it complicates comparisons to other work.

The quantification of 2-MT and 2-MGA in our work is different from Ding et al. As the reviewer mentioned that it complicates comparisons to other work. Hence, we have deleted this sentence "The higher values of 2-MTs / 2-MGA in this study than that in a previous study in Beijing at the ground level (average 1.7) (Ding et al., 2014) suggests an efficient reduction of $NO_x$ by the strict emission controls." in the revised manuscript.

P8L16: I really don't see where in Wang et al. 2005 they suggest C5-ATs are convert to 2-MTs, could the authors please point me toward that claim in that work that they are referencing? Everything I see in there is that C5-ATs are formed from IEPOX (which can also be converted to 2-MTs)

Thanks for the reviewer's comments. The claim in Wang et al. 2005 is that "C5-alkene triols hint at the formation of epoxydiol derivatives of isoprene, which can be regarded as intermediates in the formation of 2-MTs from isoprene.". We are sorry that we have misunderstood the claim of the reference. We have modified this sentence "C5-alkene triols have been suggested to convert into 2-MTs" to "Both $C_5$-alkene triols and 2-MTs can be formed from epoxydiol (IEPOX) derivatives of isoprene." in the revised manuscript. (on page 8 line 14)

P10L27-28: Given the relatively small fraction of OC accounted for the SOC tracer apportionment, do the authors have any thoughts or speculation about what the remaining portion may be? Is it all POA, or is it just due to uncertainty in the tracer apportionment? Relatedly, sesquiterpene and monoterpene SOC are presented as roughly similar - that is fairly unusual in most places, with monoterpenes usually dominating. Do the authors think this is real, or just an artifact of the tracer apportionment? Are there any other papers that have observed this?

Thanks. The small fraction of estimated SOC in OC is attributed to several reasons. The uncertainty in the tracer apportionment is one reason. Another reason is that POC should account for a large fraction of OC. We think the most important reason is that many other sources of SOC are not estimated in our study. Such as 2-methyl-3-buten-2-ol (MBO), which is emitted by pine trees, is a potential precursor of BSOA tracer (Zhang et al., 2012). Some organic sulfates and nitrates in aerosols also account for a large fraction of OC (Wang et al., 2019; Wang et al., 2021).

We think the similarities of sesquiterpene and monoterpene SOC in our study are real. A previous study mentioned that the contribution of sesquiterpene SOC is larger than monoterpene SOC in total suspended particles (TSP) over Beijing (Li et al., 2018). Another study also mentioned that the contribution of sesquiterpene and monoterpene SOC is similar in fine particles in summer over Tianjin (Fan et al., 2020).

**References**

Ding X, He QF, Shen RQ, Yu QQ, Wang XM. Spatial distributions of secondary organic aerosols from isoprene, monoterpenes, β-caryophyllene, and aromatics over China during summer. J. Geophys. Res. Atmos. 2014; 119: 1–15.

Fan Y, Liu C-Q, Li L, Ren L, Ren H, Zhang Z, et al. Large contributions of biogenic and anthropogenic sources to fine organic aerosols in Tianjin, North China. Atmospheric Chemistry and Physics 2020; 20: 117-137.

Goldstein AH, Koven CD, Heald CL, Fung IY. Biogenic carbon and anthropogenic pollutants combine to form a cooling haze over the southeastern United States. P. Natl. Acad. Sci. USA. 2009; 106: 8835–8840.

IPCC. Climate change 2014: synthesis report. Geneva, Switzerland 2014.

Li LJ, Ren LJ, Ren H, Yue SY, Xie QR, Zhao WY, et al. Molecular characterization and seasonal variation in primary and secondary organic aerosols in Beijing, China. Journal of Geophysical Research: Atmospheres 2018; 123: 12394-12412.

Wang Y, Hu M, Lin P, Tan T, Li M, Xu N, et al. Enhancement in Particulate Organic Nitrogen and Light Absorption of Humic-Like Substances over Tibetan Plateau Due to Long-Range Transported Biomass Burning Emissions. Environmental Science & Technology 2019; 53: 14222-14232.

Wang Y, Zhao Y, Wang Y, Yu J-Z, Shao J, Liu P, et al. Organosulfates in atmospheric aerosols in Shanghai, China: seasonal and interannual variability, origin, and formation mechanisms. Atmospheric Chemistry and Physics 2021; 21: 2959-2980.

Zhang H, Worton DR, Lewandowski M, Ortega J, Rubitschun CL, Park JH, et al. Organosulfates as tracers for secondary organic aerosol (SOA) formation from 2-methyl-3-buten-2-ol (MBO) in the atmosphere. Environmental Science & Technology 2012; 46: 9437-46.